# ON THE SPECTRAL BIAS OF NEURAL NETWORKS

## ABSTRACT

Neural networks are known to be a class of highly expressive functions able to fit even random input-output mappings with $100\%$ accuracy. In this work we present properties of neural networks that complement this aspect of expressivity. By using tools from Fourier analysis, we show that deep ReLU networks are biased towards low frequency functions, meaning that they cannot have local fluctuations without affecting their global behavior. Intuitively, this property is in line with the observation that over-parameterized networks find simple patterns that generalize across data samples. We also investigate how the shape of the data manifold affects expressivity by showing evidence that learning high frequencies gets *easier* with increasing manifold complexity, and present a theoretical understanding of this behavior. Finally, we study the robustness of the frequency components with respect to parameter perturbation, to develop the intuition that the parameters must be finely tuned to express high frequency functions.

## 1 INTRODUCTION

While universal approximation properties of neural networks have been known since the early 90s (Hornik et al., 1989; Cybenko, 1989; Leshno et al., 1993; Barron, 1993), recent research has shed light on the mechanisms underlying such expressivity (Montufar et al., 2014; Raghu et al., 2016; Poole et al., 2016). At the same time, deep neural networks, despite being massively over-parameterized, have been remarkably successful at generalizing to natural data. This fact is at odds with the traditional notions of model complexity and their empirically demonstrated ability to fit arbitrary random data to perfect accuracy (Zhang et al., 2017a; Arpit et al., 2017). It has prompted the recent investigations of possible implicit regularization mechanisms inherent in the learning process, inducing biases towards low complexity solutions (Soudry et al., 2017; Poggio et al., 2018; Neyshabur et al., 2017).

In this work, our main goal is to expose one such bias by taking a closer look at neural networks through the lens of Fourier analysis[1]. We focus the discussion on ReLU networks, whose piece-wise linear structure enables an analytic treatment. While they can approximate arbitrary functions, we find that these networks favour *low frequency* ones; in other words, they exhibit a bias towards smooth functions, a phenomenon we call the *spectral bias*[2]. We find that this bias manifests itself not just in the process of learning, but also in the parameterization of the model itself: in fact we show that the lower frequencies of trained networks are more robust with respect to random parameter perturbations. Finally, we also exhibit and analyze a rather intricate interplay between the spectral bias and the geometry of the data manifold: we show that high frequencies get easier to learn when the data lies on a lower dimensional manifold of complex shape embedded in the input space.

CONTRIBUTIONS

1. We exploit the piecewise-linear structure of ReLU networks to evaluate and bound its Fourier spectrum.

2. We demonstrate the peculiar behaviour of neural networks with illustrative and minimal experiments and find evidence of a *spectral bias*: i.e. lower frequencies are learned first.

---

[1] The Fourier transform affords a natural way of measuring how fast a function can change within a small neighborhood in its input of a model. See Appendix B for a brief recap of Fourier analysis.

[2] A similar result has been independently found and reported in Xu et al. (2018).

3. We illustrate how the manifold hypothesis adds a layer of subtlety by showing how the geometry of the data manifold attenuates the spectral bias in a non-trivial way. We present a theoretical analysis of this phenomenon and derive conditions on the manifolds that facilitate learning higher frequencies.

4. Given a trained network, we investigate the relative *robustness* of the lower frequencies with respect to random perturbations of the network parameters.

The paper is organized as follows. In Section 2, we derive the Fourier spectrum of deep ReLU networks. Section 3 presents minimal experiments that demonstrate the spectral bias of ReLU networks. In Section 4, we study and discuss the role of the geometry of the data manifold. In Section 5, we empirically illustrate and theoretically explain our robustness result.

## 2 FOURIER ANALYSIS OF RELU NETWORKS

### 2.1 PRELIMINARIES

Consider the class of scalar functions $f : \mathbb{R}^d \mapsto \mathbb{R}$ defined by a ReLU network with $L$ hidden layers of widths $d_1, \cdots d_L$ and a single output neuron:

$$f(\mathbf{x}) = \left( T^{(L+1)} \circ \sigma \circ T^{(L)} \circ \cdots \circ \sigma \circ T^{(1)} \right)(\mathbf{x}) \tag{1}$$

where each $T^{(k)} : \mathbb{R}^{d_{k-1}} \to \mathbb{R}^{d_k}$ is an affine function ($d_0 = d$ and $d_{L+1} = 1$) and $\sigma(\mathbf{u})_i = \max(0, u_i)$ denotes the ReLU activation function acting elementwise on a vector $\mathbf{u} = (u_1, \cdots u_n)$. In the standard basis, $T^{(k)}(\mathbf{x}) = W^{(k)}\mathbf{x} + \mathbf{b}^{(k)}$ for some weight matrix $W^{(k)}$ and bias vector $\mathbf{b}^{(k)}$.

ReLU networks are known to be continuous piece-wise linear (CPWL) functions, where the linear regions are convex polytopes (Raghu et al., 2016; Montufar et al., 2014; Zhang et al., 2018; Arora et al., 2018). Remarkably, the converse is also true: every CPWL function can be represented by a ReLU network (Arora et al., 2018, Theorem 2.1), which in turn endows ReLU networks with universal approximation properties. Given the ReLU network $f$ from Eqn. 1, we can make the piecewise linearity explicit by writing,

$$f(\mathbf{x}) = \sum_\epsilon 1_{P_\epsilon}(\mathbf{x}) \left( W_\epsilon \mathbf{x} + \mathbf{b}_\epsilon \right) \tag{2}$$

where $\epsilon$ is an index for the linear regions $P_\epsilon$ and $1_{P_\epsilon}$ is the indicator function on $P_\epsilon$. As shown in Appendix C in more detail, each region corresponds to an *activation pattern*[3] of all hidden neurons of the network, which is a binary vector with components conditioned on the sign of the input of the respective neuron. The $1 \times d$ matrix $W_\epsilon$ is given by

$$W_\epsilon = W^{(L+1)} W_\epsilon^{(L)} \cdots W_\epsilon^{(1)} \tag{3}$$

where $W_\epsilon^{(k)}$ is obtained from the original weight $W^{(k)}$ by setting its $j^{th}$ column to zero whenever the neuron $j$ of the $k^{th}$ layer is inactive.

We will henceforth assume that the input data lies in a bounded domain of $\mathbb{R}^d$, say $X = [-A, A]^d$ for some $A > 0$ and thus restrict ourselves to ReLU networks with bounded support[4].

### 2.2 FOURIER SPECTRUM

In the following, we study the structure of ReLU networks in the Fourier domain, which is defined as:

$$f(\mathbf{x}) = (2\pi)^{d/2} \int \tilde{f}(\mathbf{k}) \, e^{i\mathbf{k}\cdot\mathbf{x}} \mathbf{dk}, \qquad \tilde{f}(\mathbf{k}) := \int f(\mathbf{x}) \, e^{-i\mathbf{k}\cdot\mathbf{x}} \mathbf{dx} \tag{4}$$

---

[3]We adopt the terminology of Raghu et al. (2016); Montufar et al. (2014).

[4]Note that there is no loss of generality here: given any ReLU network $f$, its restriction $f_{|X}$ to $X$ can always be extended to a continuous piece-wise linear function on $\mathbb{R}^d$ with bounded support; this function, in turn, is representable by a ReLU network. This argument also shows that ReLU networks with bounded support are universal approximators of functions on $X$.

where $\mathbf{dx}, \mathbf{dk}$ are the uniform Lebesgue measure on $\mathbb{R}^d$ and $\tilde{f}$ denotes the Fourier transform of $f$ (see Appendix B for a short recap of the Fourier transform). Lemmas 1 and 2 (proved in appendix D) yield the explicit form of the Fourier components.

**Lemma 1.** *The Fourier transform of ReLU networks decomposes as,*

$$\tilde{f}(\mathbf{k}) = i \sum_\epsilon \frac{W_\epsilon \mathbf{k}}{k^2} \tilde{1}_{P_\epsilon}(\mathbf{k}) \tag{5}$$

*where $k = \|\mathbf{k}\|$ and $\tilde{1}_P(\mathbf{k}) = \int_P e^{-i\mathbf{k}\cdot\mathbf{x}}\mathbf{dx}$ is the Fourier transform of the indicator function of $P$.*

The Fourier transform of a polytope appearing in Eqn. 5 is a fairly intricate mathematical object; Diaz et al. (2016) develop an elegant procedure for evaluating it in arbitrary dimensions via a recursive application of Stokes theorem. We describe this procedure in detail in Appendix D.2, and present here its main corollary.

**Lemma 2.** *Let $P$ be a full dimensional polytope in $\mathbb{R}^d$. The Fourier spectrum of its indicator function $\tilde{1}_P$ satisfies the following:*

$$|\tilde{1}_P(\mathbf{k})| = \mathcal{O}\left(\frac{1}{k^{\Delta_\mathbf{k}^{(P)}}}\right) \tag{6}$$

*where $1 \leq \Delta_\mathbf{k}^{(P)} \leq d$, and $\Delta_\mathbf{k}^{(P)} = j$ when $\mathbf{k}$ lies orthogonal to some $(d-j)$-dimensional face of $P$.*

Note that since a polytope has a finite number of facets (of any dimension), the $\mathbf{k}$'s for which $\Delta_\mathbf{k}^{(P)} = j$ for some $j < d$ lie on a finite union of $j$-dimensional subspaces of $\mathbb{R}^d$. The Lebesgue measure of all such lower dimensional subspaces for all such $j$ equals 0, leading us to the conclusion that the spectrum decays as $\mathcal{O}(k^{-d})$ for *almost all* directions $\hat{\mathbf{k}}$ of the frequency vector $\mathbf{k}$ in $\mathbb{R}^d$.

Lemmas 1, 2 together yield the main result of this section. Given a ReLU network $f$, its linear regions form a cell decomposition of $\mathbb{R}^d$ as union of polytopes; we denote by $\mathcal{F}$ the set of faces (of any dimension) of all such polytopes. For $\mathbf{k} \in \mathbb{R}^d$, let $\Delta_\mathbf{k}$ be the minimum integer $1 \leq j \leq d$ such that $\mathbf{k}$ lies orthogonal to some $(d-j)$-dimensional face in $\mathcal{F}$.

**Theorem 1.** *The Fourier components of the ReLU network $f$ satisfy the following:*

$$|\tilde{f}(\mathbf{k})| = \mathcal{O}\left(\frac{N_f L_f}{k^{\Delta_\mathbf{k}+1}}\right) \tag{7}$$

*where $N_f$ is the number of linear regions and $L_f = \max_\epsilon \|W_\epsilon\|_2$ is the Lipschitz constant of $f$.*

Several remarks are in order:

(a) The spectral decay of ReLU networks is highly anisotropic in large dimensions. In almost all directions $\hat{\mathbf{k}}$ of $\mathbb{R}^d$, we have $\Delta_\mathbf{k} = d$, i.e. a $\mathcal{O}(k^{-d-1})$ decay. However, the decay can be as slow as $\mathcal{O}(k^{-2})$ in specific directions orthogonal to the facets bounding linear regions[5].

As we prove in Appendix D.3, the Lipschitz constant $L_f$ can be bounded as,

$$L_f \leq \prod_{k=1}^{L+1} \|W^{(k)}\| \leq \|W\|_\infty^{L+1} \sqrt{d} \prod_{k=1}^{L} d_k \tag{8}$$

where $\|\cdot\|$ is the spectral norm, $W$ is the ravelled parameter vector of the network and $\|\cdot\|_\infty$ is the max-norm. This makes the bound on $L_f$ scale exponentially in depth and polynomial in width. As for the number $N_f$ of linear regions, Montufar et al. (2014) and Raghu et al. (2016) obtain tight bounds that exhibit the same scaling behaviour (Raghu et al., 2016, Theorem 1). This makes the overall bound in Eqn. 7 – and with it, the ability to express larger frequencies – scale exponentially in depth and polynomially in width[6]. This result complements the well-known universal approximation property of neural networks by explicitly incorporating a control on the capacity[7] of the network, namely the width, depth and the norm of parameters. Architecture dependent controls

---

[5]Note that such a rate is *not* guaranteed by piecewise smoothness alone. For instance, the function $\sqrt{|x|}$ is continuous and smooth everywhere except at $x = 0$, yet it decays as $k^{-1.5}$ in the Fourier domain.

[6]A qualitative ablation study can be found in Appendix A.3.

[7]Note that the bound is relaxed with increasing capacity. In the limit of capacity to infinity, any constraint on universal approximation is dissolved.

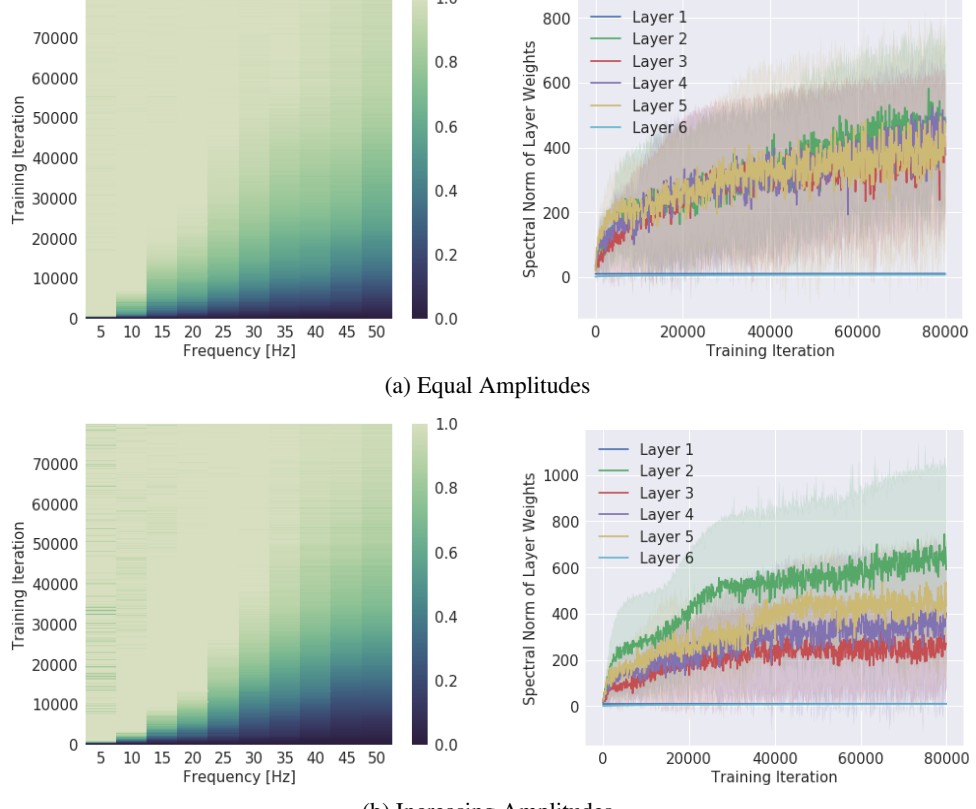

(a) Equal Amplitudes

(b) Increasing Amplitudes

Figure 1: Left (a, b): Evolution of the spectrum (x-axis for frequency) during training (y-axis). The colors show the measured amplitude of the network spectrum at the corresponding frequency, normalized by the target amplitude at the same frequency (i.e. $|\tilde{f}_{k_i}|/A_i$) and the colorbar is clipped between 0 and 1. Right (a, b): Evolution of the spectral norm (y-axis) of each layer during training (x-axis). Figure-set (a) shows the setting where all frequency components in the target function have the same amplitude, and (b) where higher frequencies have larger amplitudes. **Gist**: We find that even when higher frequencies have larger amplitudes, the model prioritizes learning lower frequencies first. We also find that the spectral norm of weights increases as the model fits higher frequency, which is what we expect from Theorem 1.

on approximation have been formalized in the literature through approximation bounds and depth separation results, see e.g Barron (1993); Telgarsky (2016); Eldan & Shamir (2016).

(b) For a given architecture (i.e. fixed width and depth), the high frequency contributions of the network can be increased by increasing the norm of the parameters. Assuming the weight norm increases with training iterations, this suggests that the training of ReLU networks might be biased towards lower frequencies. We investigate this fact empirically in the next section.

## 3 LOWER FREQUENCIES ARE LEARNED FIRST

In this section and the ones that follow, we present experiments that illustrate the peculiar behaviour of deep ReLU networks in the Fourier domain. We begin with an experiment to demonstrate that networks tend to fit *lower frequencies first* during training. We refer to this phenomenon as the *spectral bias*, and discuss it in light of the results of Section 2.

**Experiment 1.** The setup is as follows[8] : Given frequencies $\kappa = (k_1, k_2, ...)$ with corresponding amplitudes $\alpha = (A_1, A_2, ...)$, and phases $\phi = (\varphi_1, \varphi_2, ...)$, we consider the mapping $\lambda : [0, 1] \to \mathbb{R}$ given by

$$\lambda(z) = \sum_i A_i \sin(2\pi k_i z + \varphi_i). \tag{9}$$

---

[8]More experimental details and additional plots are provided in Appendix A.1.

A 6-layer deep 256-unit wide ReLU network $f_\theta$ is trained to regress $\lambda$ with $\kappa = (5, 10, ..., 45, 50)$ and $N = 200$ input samples spaced equally over $[0, 1]$; its spectrum $\tilde{f}_\theta(k)$ in expectation over $\varphi_i \sim U(0, 2\pi)$ is monitored as training progresses. In the first setting, we set equal amplitude $A_i = 1$ for all frequencies and in the second setting, the amplitude increases from $A_1 = 0.1$ to $A_{10} = 1$. Fig 1 shows the normalized magnitudes $|\tilde{f}_\theta(k_i)|/A_i$ at various frequencies, as training progresses. The result is that lower frequencies (i.e. smaller $k_i$'s) are regressed first, regardless of their amplitudes.

**Discussion.** Multiple theoretical aspects may underlie these observations. First, for a fixed architecture, the bound in Theorem 1 allows for larger Fourier coefficients at higher frequencies if the parameter norm is large. However, the parameter norm can increase only gradually during training by gradient descent, which leads to the higher frequencies being learned late in the optimization process. To confirm that the bound indeed increases as the model fits higher frequencies, we plot in Fig 1 the spectral norm of weights of each layer during training for both cases of constant and increasing amplitudes.

Second, consider the Mean Squared Error $\text{MSE}[f_\theta, \lambda]$ in terms of the Fourier components: letting $z_i = i/N$ be the training sample points, we have:

$$\text{MSE}[f_\theta, \lambda] = \frac{1}{N} \sum_{i=0}^{N-1} |f_\theta(z_i) - \lambda(z_i)|^2 = \frac{1}{N} \sum_{k=0}^{N-1} |\tilde{f}_\theta(k) - \tilde{\lambda}(k)|^2 = \text{MSE}[\tilde{f}_\theta, \tilde{\lambda}] \qquad (10)$$

where the second equality follows from Plancherel theorem. We make two observations – first, the square error in input space translates into square error in Fourier domain, with a priori no structural bias towards any particular frequency component[9], i.e. all frequencies are weighted the same. Since the same cannot be said about e.g. cross-entropy loss, we use the MSE loss in most of our experiments to avoid a potential confounding factor. Second, the parameterization of the network can be exploited by considering the gradient of the MSE loss w.r.t. parameters,

$$\frac{\partial}{\partial \theta} \text{MSE}[\tilde{f}_\theta, \tilde{\lambda}] = \frac{2}{N} \sum_{k=0}^{N-1} \text{Re}[\tilde{f}_\theta(k) - \tilde{\lambda}(k)] \frac{\partial \tilde{f}_\theta(k)}{\partial \theta} \leq \frac{2}{N} \sum_{k=0}^{N-1} |\tilde{f}_\theta(k) - \tilde{\lambda}(k)| \underbrace{\left| \frac{\partial \tilde{f}_\theta(k)}{\partial \theta} \right|}_{=\mathcal{O}(k^{-\Delta-1})} \qquad (11)$$

where $\text{Re}(z)$ denotes the real part of $z$. We find that a bias naturally emerges as a consequence of the spectral decay rate found in Theorem 1, in the sense that the magnitude of the residual $|\tilde{f}_\theta(k) - \tilde{\lambda}(k)|$ contributes less to the net gradient for large $k$. This generalizes the argument made in Xu (2018) for two layer sigmoid networks by observing that the gradient w.r.t parameters of the network function inherits the spectral decay rate of the function itself[10]. In Section 5, we use that the integral of $\tilde{f}_\theta$ w.r.t. the standard measure in parameter space $d\theta$ also inherits the spectral decay rate of $\tilde{f}$ to make a statement about the robustness of $\tilde{f}_\theta(k)$ against random parameter perturbations.

## 4 Not all Manifolds are Learned Equal

In this section, we investigate the subtleties that arise when the data lies on a lower dimensional manifold embedded in the higher dimensional input space of the model (Goodfellow et al., 2016). We find that the *shape* of the data-manifold impacts the learnability of high frequencies in a non-trivial way. As we shall see, this is because low frequencies functions in the input space may have high frequency components when restricted to lower dimensional manifolds of complex shapes. To systematically investigate the impact of manifold shape on the spectral bias, we demonstrate results in an illustrative minimal setting[11] free from unwanted confounding factors. We also present a mathematical exposition of the relationship between the Fourier spectrum of the network, the spectrum of the target function defined on the manifold, and the geometry of the manifold itself.

---

[9]Note that the finite range in frequencies is due to sampling

[10]Observe that the partial derivative w.r.t. $\theta$ can be swapped with the integrals in Eqn 4.

[11]We include experiments on MNIST and CIFAR-10 in appendices A.4 and A.5.

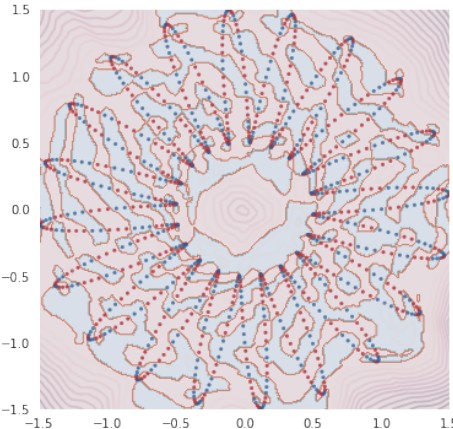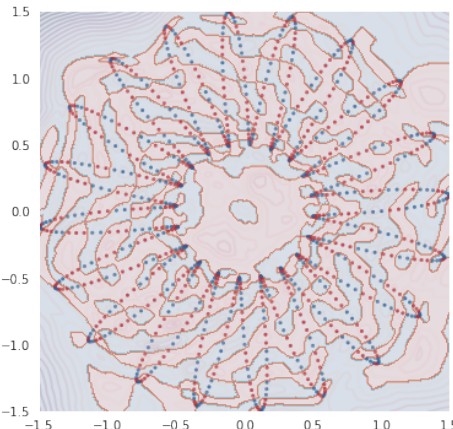

Figure 2: Functions learned by two identical networks (up to initialization) to classify the binarized value of a sine wave of frequency $k = 200$ defined on a $\gamma_{L=20}$ manifold. Both yield close to perfect accuracy for the samples defined on the manifold (scatter plot), yet they differ significantly elsewhere. The shaded regions show the predicted class (Red or Blue) whereas contours show the confidence (absolute value of logits).

**Manifold hypothesis.** We consider the case where the data lies on a lower dimensional *data manifold* $\mathcal{M} \subset \mathbb{R}^d$ embedded in input space, which we assume to be the image $\gamma([0,1]^m)$ of some injective mapping $\gamma : [0,1]^m \to \mathbb{R}^d$ defined on a lower dimensional latent space $[0,1]^m$. Under this hypothesis and in the context of the standard regression problem, a target function $\tau : \mathcal{M} \to \mathbb{R}$ defined on data manifold can identified with a function $\lambda = \tau \circ \gamma$ defined on the latent space. Regressing $\tau$ is therefore equivalent to finding $f : \mathbb{R}^d \to \mathbb{R}$ such that $f \circ \gamma$ matches $\lambda$. Further, assuming that the data probability distribution $\mu$ supported on $\mathcal{M}$ is induced by $\gamma$ from the uniform distribution $U$ in the latent space $[0,1]^m$, the mean square error can be expressed as,

$$\text{MSE}_\mu^{(\mathbf{x})}[f, \tau] = \mathbb{E}_{\mathbf{x} \sim \mu} |f(\mathbf{x}) - \tau(\mathbf{x})|^2 = \mathbb{E}_{\mathbf{z} \sim U} |(f(\gamma(\mathbf{z})) - \lambda(\mathbf{z})|^2 = \text{MSE}_U^{(\mathbf{z})}[f \circ \gamma, \lambda] \quad (12)$$

Observe that there is a vast space of degenerate solutions $f$ that minimize the mean squared error – namely all functions on $\mathbb{R}^d$ that yield the same function when restricted to the data manifold $\mathcal{M}$.

Our findings from the previous section suggest that neural networks are biased towards expressing a particular subset of such solutions, namely those that are low frequency. It is also worth noting that there exist methods that restrict the space of solutions: notably adversarial training (Goodfellow et al., 2014) and Mixup (Zhang et al., 2017b).

**Experimental set up.** The experimental setting is designed to afford control over both the shape of the data manifold and the target function defined on it. We will consider the family of curves in $\mathbb{R}^2$ generated by mappings $\gamma_L : [0,1] \to \mathbb{R}^2$ given by

$$\gamma_L(z) = R_L(z)(\cos(2\pi z), \sin(2\pi z)) \text{ where } R_L(z) = 1 + \frac{1}{2}\sin(2\pi L z) \quad (13)$$

Here, $\gamma_L([0,1])$ defines the data-manifold and corresponds to a flower-shaped curve with $L$ petals, or a unit circle when $L = 0$ (see e.g. Fig 2). Given a signal $\lambda : [0,1] \to \mathbb{R}$ defined on the latent space $[0,1]$, the task entails learning a network $f : \mathbb{R}^2 \to \mathbb{R}$ such that $f \circ \gamma_L$ matches the signal $\lambda$.

**Experiment 2.** The set-up is similar to that of Experiment 1, and $\lambda$ is as defined in Eqn. 9 with frequencies $\kappa = (20, 40, ..., 180, 200)$, and amplitudes $A_i = 1 \forall i$. The model $f$ is trained on the dataset $\{\gamma_L(z_i), \lambda(z_i)\}_{i=1}^N$ with $N = 1000$ uniformly spaced samples $z_i$ between 0 and 1. The spectrum of $f \circ \gamma_L$ in expectation over $\varphi_i \sim U(0, 2\pi)$ is monitored as training progresses, and the result shown in Fig 3 for $L = 0, 4, 10, 16$. Fig 3e shows the corresponding mean squared error curves. More experimental details in appendix A.2.

The results demonstrate a clear attenuation of the spectral bias as $L$ grows. Moreover, Fig 3e suggests that the larger the $L$, the easier the learning task.

**Experiment 3.** Here, we adapt the setting of Experiment 2 to binary classification by simply thresholding the function $\lambda$ at 0.5 to obtain a binary target signal. To simplify visualization, we only use

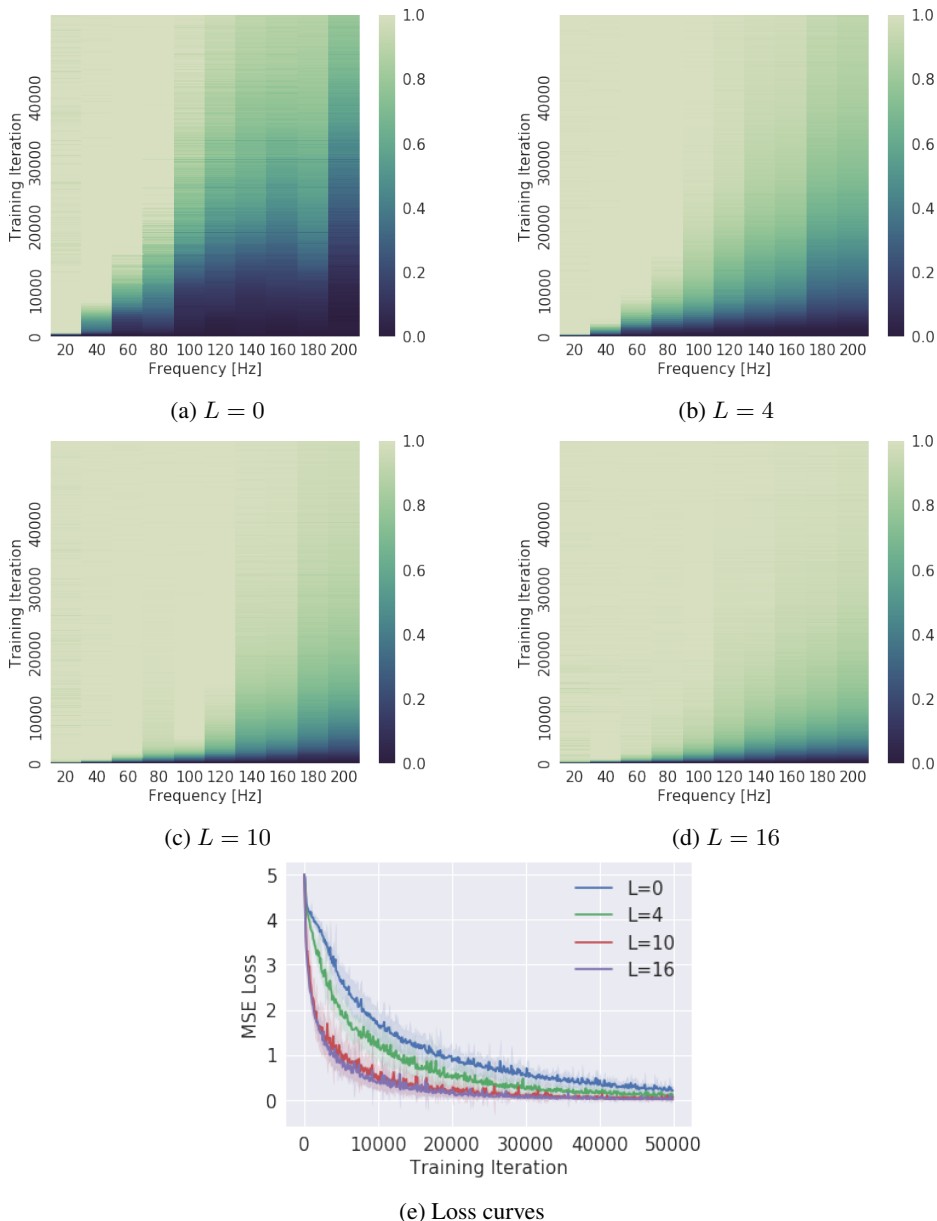

(a) $L = 0$

(b) $L = 4$

(c) $L = 10$

(d) $L = 16$

(e) Loss curves

Figure 3: (a,b,c,d): Evolution of the network spectrum (x-axis for frequency, colorbar for magnitude) during training (y-axis) for the same target functions defined on manifolds $\gamma_L$ for various $L$. Since the target function has amplitudes $A_i = 1$ for all frequencies $k_i$ plotted, the colorbar is clipped between 0 and 1. (e): Corresponding learning curves. **Gist**: Some manifolds (here with larger $L$) make it easier for the network to learn higher frequencies than others.

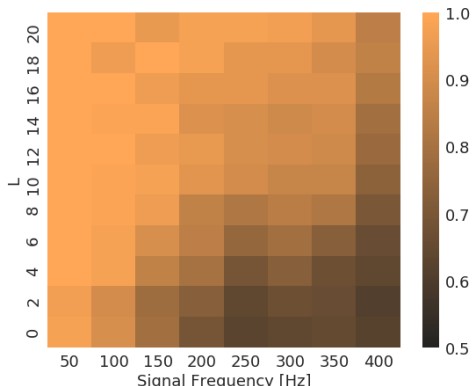

Figure 4: Heatmap of training accuracies of a network trained to predict the binarized value of a sine wave of given frequency (x-axis) defined on $\gamma_L$ for various $L$ (y-axis).

signals with a single frequency mode $k$, such that $\lambda(z) = \sin(2\pi k z + \varphi)$. We train the same network on the resulting classification task with cross-entropy loss[12] for $k \in \{50, 100, ..., 350, 400\}$ and $L \in \{0, 2, ..., 18, 20\}$. The heatmap in Fig 4 shows the classification accuracy for each $(k, L)$ pair. Fig 2 shows visualizations of the functions learned by the same network, trained on $(k, L) = (200, 20)$ under identical conditions up to random initialization.

Observe that increasing $L$ (i.e. going up a column in Fig 4) results in better (classification) performance for the same target signal. This is the same behaviour as we observed in Experiment 2 (Fig 3a-d), but now with binary cross-entropy loss instead of the MSE.

**Discussion.** These experiments hint towards a rich interaction between the shape of the manifold and the effective difficulty of the learning task. The key technical reason underlying this phenomenon (as we formalize below) is that the relationship between frequency spectrum of the network $f$ and that of the fit $f \circ \gamma_L$ is mediated by the embedding map $\gamma_L$. In particular, we will argue that a given signal defined on the manifold is easier to fit when the coordinate functions of the manifold embedding itself has high frequency components. Thus, in our experimental setting, the same signal embedded in a flower with more petals can be captured with lower frequencies of the network.

To understand this mathematically, we address the following questions: given a target function $\lambda$, how small can the frequencies of a solution $f$ be such that $f \circ \gamma = \lambda$? And further, how does this relate to the geometry of the data-manifold $\mathcal{M}$ induced by $\gamma$? To find out, we write the Fourier transform of the composite function,

$$\widetilde{(f \circ \gamma)}(\mathbf{l}) = \int \mathbf{dk} \tilde{f}(\mathbf{k}) P_\gamma(\mathbf{l}, \mathbf{k}) \quad \text{where} \quad P_\gamma(\mathbf{l}, \mathbf{k}) = \int_{[0,1]^m} \mathbf{dz}\, e^{i(\mathbf{k} \cdot \gamma(\mathbf{z}) - \mathbf{l} \cdot \mathbf{z})} \tag{14}$$

The kernel $P_\gamma$ depends on only $\gamma$ and elegantly encodes the correspondence between frequencies $\mathbf{k} \in \mathbb{R}^d$ in input space and frequencies $\mathbf{l} \in \mathbb{R}^m$ in the latent space $[0, 1]^m$. Following a procedure from Bergner et al., we can further investigate the behaviour of the kernel in the regime where the stationary phase approximation is applicable, i.e. when $l^2 + k^2 \to \infty$ (cf. section 3.2. of Bergner et al.). In this regime, the integral $P_\gamma$ is dominated by critical points $\bar{\mathbf{z}}$ of its phase, which satisfy

$$\mathbf{l} = J_\gamma(\bar{\mathbf{z}})\, \mathbf{k} \tag{15}$$

where $J_\gamma(\mathbf{z})_{ij} = \nabla_i \gamma_j(\mathbf{z})$ is the $m \times d$ Jacobian matrix of $\gamma$. Non-zero values of the kernel correspond to pairs $(\mathbf{l}, \mathbf{k})$ such that Eqn 15 has a solution. Further, given that the components of $\gamma$ (i.e. its coordinate functions) are defined on an interval $[0, 1]^m$, one can use their Fourier series representation together with Eqn 15 to obtain a condition on their frequencies (shown in appendix D.4). More precisely, we find that the $i$-th component of the RHS in Eqn 15 is proportional to $\mathbf{p} \tilde{\gamma}_i[\mathbf{p}] k_i$ where $\mathbf{p} \in \mathbb{Z}^m$ is the frequency of the coordinate function $\gamma_i$. This yields that we can get arbitrarily large frequencies $l_i$ if $\tilde{\gamma}_i[\mathbf{p}]$ is large[13] enough for large $\mathbf{p}$, even when $k_i$ is fixed.

---

[12]We use Pytorch's `BCEWithLogitsLoss`. Internally, it takes a sigmoid of the network's output (the logits) before evaluating the cross-entropy.

[13]Consider that the data-domain is bounded, implying that $\tilde{\gamma}$ cannot be arbitrarily scaled.

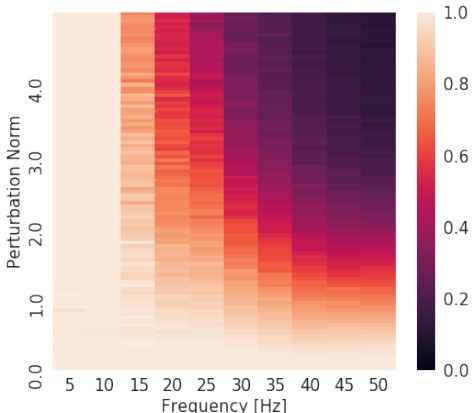

Figure 5: Normalized spectrum of the model (x-axis for frequency, colorbar for magnitude) with perturbed parameters as a function of parameter perturbation (y-axis). The colormap is clipped between 0 and 1. Observe that the lower frequencies are more robust to parameter perturbations than the higher frequencies.

This is precisely what Experiments 2 and 3 demonstrate in a minimal setting. From Eqn 13, observe that the coordinate functions have a frequency mode at $L$. For increasing $L$, it is apparent that the frequency magnitudes $l$ (in the latent space) that can be expressed with the same frequency $k$ (in the input space) increases with increasing $L$. This allows the remarkable interpretation that the neural network function can express large frequencies on a manifold ($l$) with smaller frequencies w.r.t its input domain ($k$), provided that the coordinate functions of the data manifold embedding itself has high-frequency components[14].

## 5  LOWER FREQUENCIES ARE MORE ROBUST

The goal of this section is to show that lower frequency components of trained networks are more *robust* than their higher frequency counterparts with respect to random perturbations in parameter space. More precisely, we observe that in the neighbourhood of a solution in parameter space, the high frequency components decay faster than the low frequency ones. This property does not directly depend on the training process, but rather on the parametrization of the trained model. We present empirical evidence and a theoretical explanation of this phenomenon.

**Experiment 4.** The set up is the same as in Experiment 1, where $\lambda$ is given by Eqn. 9. Training is performed for the frequencies $\kappa = (10, 15, 20, ..., 45, 50)$ and amplitudes $A_i = 1 \forall i$. After convergence to $\theta^*$, we consider random (isotropic) perturbations $\theta = \theta^* + \delta\hat{\theta}$ of given magnitude $\delta$, where $\hat{\theta} \sim U(\mathcal{S}^{\dim(\theta^*)})$ is a unit vector. We evaluate the network function $f_\theta$ at the perturbed parameters, and compute the magnitude of its discrete Fourier transform at frequencies $k_i$, $|\tilde{f}_\theta(k_i)|$. We also average over 100 samples of $\hat{\theta}$ to obtain $|\tilde{f}_{\mathbb{E}\theta}(k_i)|$, which we normalize by $|\tilde{f}_{\theta*}(k_i)|$. The result, shown in Figure 5, demonstrate that higher frequencies are significantly less robust than the lower ones.

**Discussion.** The interpretation is as follows: parameters that contribute towards expressing high-frequency components occupy a small volume in the parameter space. To formalize this intuition, given a bounded domain $\Theta$ of parameter space, let us define,

$$\Xi_\epsilon(k) = \{\theta \in \Theta | \exists \mathbf{k}', k' > k, |\tilde{f}_\theta(\mathbf{k}')| > \epsilon\}$$

to be the set of parameters such that $f_\theta$ has Fourier components larger than $\epsilon$ for some $\mathbf{k}'$ with larger norm than $k$. Then the following Proposition holds (proved in appendix E).

**Proposition 1.** *The volume ratio,*

$$R(k) = \frac{Vol(\Xi_\epsilon(k))}{Vol(\Theta)} \tag{16}$$

---

[14]Informally, we allow ourselves the intuition that *it's easy for neural networks to fit wiggly functions if they are defined on wiggly manifolds.*

*inherits the spectral decay rate of $|\tilde{f}_\theta(\mathbf{k})|$, given by Theorem 1.*

Intuitively, expressing larger frequencies requires the parameters to be finely-tuned to work together.

## 6 RELATED WORK

While we focus on showing the spectral bias of deep ReLU networks towards learning functions with dominant lower frequency components, most of existing work has focused on showing that in theory, these networks are capable of learning arbitrarily complex functions. Hornik et al. (1989); Cybenko (1989); Leshno et al. (1993) have shown that neural networks can be universal approximators when given sufficient width; more recently, Lu et al. (2017) proved that this property holds also for width-bounded networks. Montufar et al. (2014) showed that the number of linear regions of deep ReLU networks grows polynomially with width and exponentially with depth; Raghu et al. (2016) generalized this result and provided asymptotically tight bounds. There have been various results of the benefits of depth for efficient approximation (Poole et al., 2016; Telgarsky, 2016; Eldan & Shamir, 2016). These analysis on the expressive power of deep neural networks can in part explain why over-parameterized networks can perfectly learn random input-output mappings (Zhang et al., 2017a). Our Fourier analysis of deep ReLU networks also reflects the width and depth dependence of their expressivity, but more interestingly reveals their spectral bias towards learning simple functions. Thus our work may be seen as a formalization of the findings of Arpit et al. (2017), where it is empirically shown that deep networks prioritize learning simple functions during training.

A few other works studied neural networks through the lens of harmonic analysis. For example, Candès (1999) used the ridgelet transform to build constructive procedures for approximating a given function by neural networks, in the case of oscillatory activation functions. This approach has been recently generalized to unbounded activation functions by Sonoda & Murata (2017). Eldan & Shamir (2016) use insights on the support of the Fourier spectrum of two-layer networks to derive a worse-case depth-separation result. Barron (1993) makes use of Fourier space properties of the target function to derive an architecture-dependent approximation bound. In a work done independently from ours, and made available online almost at the same time, Xu et al. (2018) make the same observation that lower frequencies are learned first. The subsequent work by Xu (2018) proposes a theoretical analysis of the phenomenon in the case of 2-layer networks with sigmoid activation, based on the spectrum of the sigmoid function.

In light of our findings, it is worth comparing the case of neural networks and other popular algorithms such that kernel machines (KM) and $K$-nearest neighbor classifiers. We refer to the Appendix F for a detailed discussion and references. In summary, our discussion there suggests that 1. DNNs strike a good balance between function smoothness and expressivity/parameter-efficiency compared with KM; 2. DNNs learn a smoother function compared with $K$NNs since the spectrum of the DNN decays faster compared with $K$NNs in the experiments shown there.

## 7 CONCLUSION

We studied deep ReLU networks through the lens of Fourier analysis. Several conclusions can be drawn from our analysis. While neural networks can approximate arbitrary functions, we find that they favour *low frequency* ones – hence they exhibit a bias towards smooth functions – a phenomenon that we called *spectral bias*. We also illustrated how the geometry of the data manifold impacts expressivity in a non-trivial way, as high frequency functions defined on complex manifolds can be expressed by lower frequency network functions defined in input space. Finally, we found that the parameters contributing towards expressing lower frequencies are more robust to random perturbations than their higher frequency counterparts.

We view future work that explore the properties of neural networks in Fourier domain as promising. For example, the Fourier transform affords a natural way of measuring how fast a function can change within a small neighborhood in its input domain ; as such, it is a strong candidate for quantifying and analyzing the *sensitivity* of a model – which in turn provides a natural measure of complexity (Novak et al., 2018). We hope to encourage more research in this direction.

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

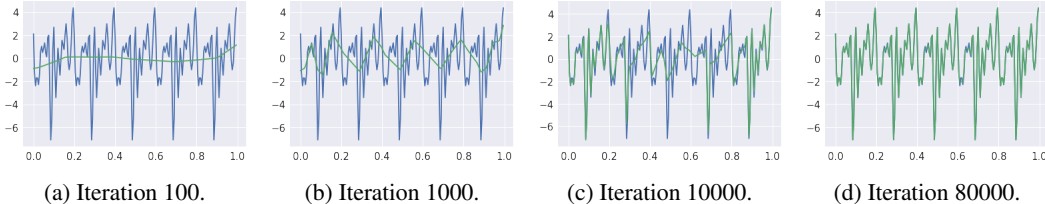

| (a) Iteration 100. | (b) Iteration 1000. | (c) Iteration 10000. | (d) Iteration 80000. |

Figure 6: The learnt function (green) overlayed on the target function (blue) as the training progresses. The target function is a superposition of sinusoids of frequencies $\kappa = (5, 10, ..., 45, 50)$, equal amplitudes and randomly sampled phases.

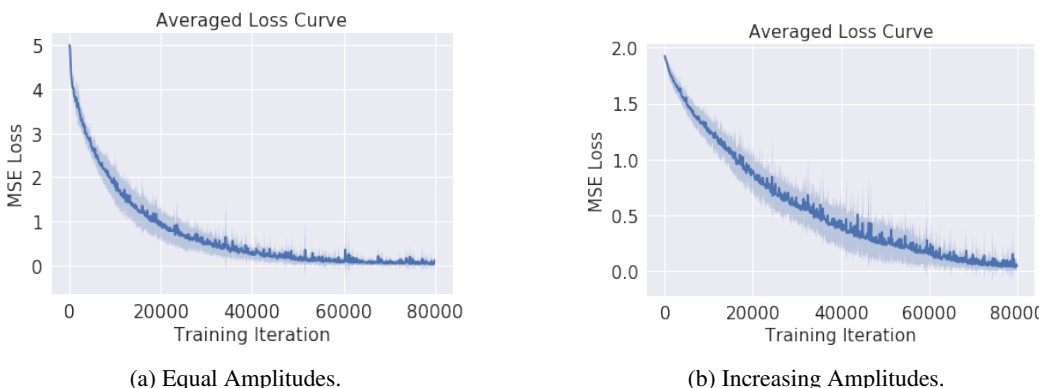

(a) Equal Amplitudes.          (b) Increasing Amplitudes.

Figure 7: Loss curves averaged over multiple runs. (cf. Experiment 1)

# A    EXPERIMENTAL DETAILS

## A.1    EXPERIMENT 1

We fit a 6 layer ReLU network with 256 units per layer $f_\theta$ to the target function $\lambda$, which is a superposition of sine waves with increasing frequencies:

$$\lambda : [0, 1] \to \mathbb{R}, \ \lambda(z) = \sum_i A_i \sin(2\pi k_i z + \varphi_i)$$

where $k_i = (5, 10, 15, ..., 50)$, and $\varphi_i$ is sampled from the uniform distribution $U(0, 2\pi)$. In the first setting, we set equal amplitude for all frequencies, i.e. $A_i = 1 \forall i$, while in the second setting we assign larger amplitudes to the higher frequencies, i.e. $A_i = (0.1, 0.2, ..., 1)$. We sample $\lambda$ on 200 uniformly spaced points in $[0, 1]$ and train the network for 80000 steps of full-batch gradient descent with Adam (Kingma & Ba, 2014). Note that we do not use stochastic gradient descent to avoid the stochasticity in parameter updates as a confounding factor. We evaluate the network on the same 200 point grid every 100 training steps and compute the magnitude of its (single-sided) discrete fourier transform at frequencies $k_i$ which we denote with $|\tilde{f}_{k_i}|$. Finally, we plot in figure 1 the normalized magnitudes $\frac{|\tilde{f}_{k_i}|}{A_i}$ averaged over 10 runs (with different sets of sampled phases $\varphi_i$). We also record the spectral norms of the weights at each layer as the training progresses, which we plot in figure 1 for both settings (the spectral norm is evaluated with 10 power iterations). In figure 6, we show an example target function and the predictions of the network trained on it (over the iterations), and in figure 7 we plot the loss curves.

## A.2    EXPERIMENT 2

We use the same 6-layer deep 256-unit wide network and define the target function

$$\lambda : \mathcal{D} \to \mathbb{R}, \ z \mapsto \lambda(z) = \sum_i A_i \sin(2\pi k_i z + \varphi_i)$$

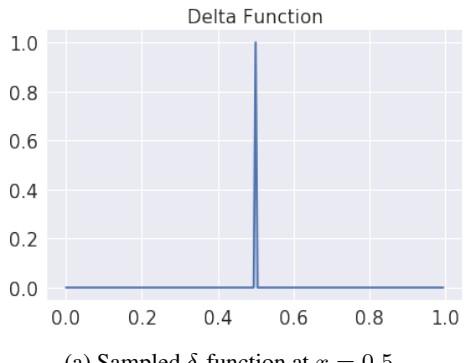
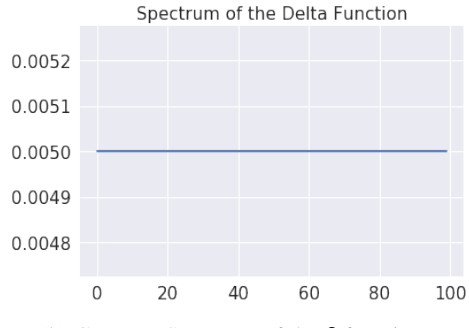

(a) Sampled $\delta$-function at $x = 0.5$.  (b) Constant Spectrum of the $\delta$-function.

Figure 8: The target function used in Experiment 5.

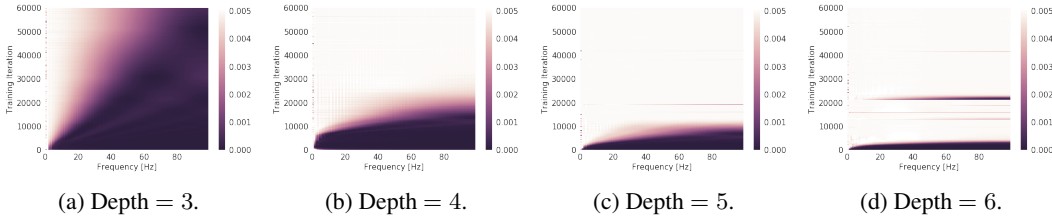

(a) Depth $= 3$.  (b) Depth $= 4$.  (c) Depth $= 5$.  (d) Depth $= 6$.

Figure 9: Evolution with training iterations (y-axis) of the Fourier spectrum (x-axis for frequency, and colormap for magnitude) for a network with **varying depth**, width $= 16$ and weight clip $= 10$. The spectrum of the target function is a constant 0.005 for all frequencies.

where $k_i = (20, 40, ..., 180, 200)$, $A_i = 1 \, \forall \, i$ and $\varphi \sim U(0, 2\pi)$. We sample $\phi$ on a grid with 1000 uniformly spaced points between 0 and 1 and map it to the input domain via $\gamma_L$ to obtain a dataset $\{(\gamma_L(z_j), \lambda(z_j))\}_{j=0}^{999}$, on which we train the network with 50000 full-batch gradient descent steps of Adam. On the same 1000-point grid, we evaluate the magnitude of the (single-sided) discrete Fourier transform of $f_\theta \circ \gamma_L$ every 100 training steps at frequencies $k_i$ and average over 10 runs (each with a different set of sampled $z_i$'s). Fig 3 shows the evolution of the spectrum as training progresses for $L = 0, 4, 10, 16$, and Fig 3e shows the corresponding loss curves.

### A.3  QUALITATIVE ABLATION OVER ARCHITECTURES

Theorem 1 exposes the relationship between the fourier spectrum of a network and its depth, width and max-norm of parameters. The following experiment is a qualitative ablation study over these variables.

**Experiment 5.** In this experiment, we fit various networks to the $\delta$-function at $x = 0.5$ (see Fig 8a). Its spectrum is constant for all frequencies (Fig 8b), which makes it particularly useful for testing how well a given network can fit large frequencies. Fig 11 shows the ablation over weight clip (i.e. max parameter max-norm), Fig 9 over depth and Fig 10 over width. Fig 12 exemplarily shows how the network prediction evolves with training iterations. All networks are trained for 60K iterations of full-batch gradient descent under identical conditions (Adam optimizer with $lr = 0.0003$, no weight decay).

We make the following observations.

(a) Fig 9 shows that increasing the depth (for fixed width) significantly improves the network's ability to fit higher frequencies (note that the depth increases linearly).

(b) Fig 10 shows that increasing the width (for fixed depth) also helps, but the effect is considerably weaker (note that the width increases exponentially).

(c) Fig 11 shows that increasing the weight clip (or the max parameter max-norm) also helps the network fit higher frequencies.

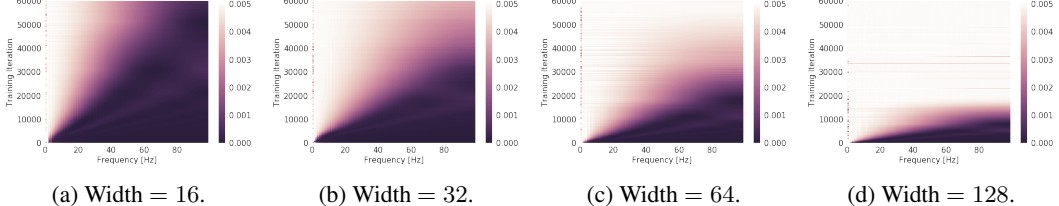

(a) Width $= 16$.  (b) Width $= 32$.  (c) Width $= 64$.  (d) Width $= 128$.

Figure 10: Evolution with training iterations (y-axis) of the Fourier spectrum (x-axis for frequency, and colormap for magnitude) for a network with **varying width**, depth $= 3$ and weight clip $= 10$. The spectrum of the target function is a constant 0.005 for all frequencies.

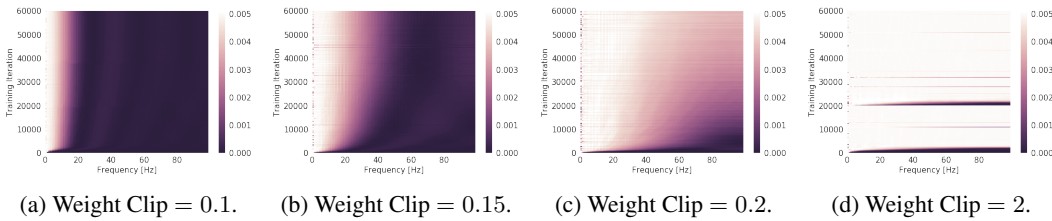

(a) Weight Clip $= 0.1$.  (b) Weight Clip $= 0.15$.  (c) Weight Clip $= 0.2$.  (d) Weight Clip $= 2$.

Figure 11: Evolution with training iterations (y-axis) of the Fourier spectrum (x-axis for frequency, and colormap for magnitude) for a network with **varying weight clip**, depth $= 6$ and width $= 64$. The spectrum of the target function is a constant 0.005 for all frequencies.

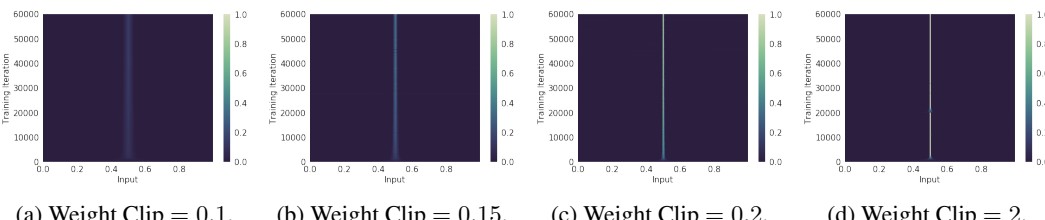

(a) Weight Clip $= 0.1$.  (b) Weight Clip $= 0.15$.  (c) Weight Clip $= 0.2$.  (d) Weight Clip $= 2$.

Figure 12: Evolution with training iterations (y-axis) of the network prediction (x-axis for input, and colormap for predicted value) for a network with **varying weight clip**, depth $= 6$ and width $= 64$. The target function is a $\delta$ peak at $x = 0.5$.

The above observations are all consistent with Theorem 1, and further show that lower frequencies are learned first (i.e. the spectral bias, cf. Experiment 1).

## A.4 MNIST: A PROOF OF CONCEPT

In the following experiment, we show that given two manifolds of the same dimension – one flat and the other not – the task of learning random labels is harder to solve if the input samples lie on the same manifold. We demonstrate on MNIST under the assumption that the manifold hypothesis is true, and use the fact that the spectrum of the target function we use (white noise) is constant in expectation, and therefore independent of the underlying coordinate system when defined on the manifold.

**Experiment 6.** In this experiment, we investigate if it is easier to learn a signal on a more realistic data-manifold like that of MNIST (assuming the manifold hypothesis is true), and compare with a flat manifold of the same dimension. To that end, we use the $64$-dimensional feature-space $\mathcal{E}$ of a denoising[15] autoencoder as a proxy for the real data-manifold of unknown number of dimensions. The decoder functions as an embedding of $\mathcal{E}$ in the input space $X = \mathbb{R}^{784}$, which effectively amounts to training a network on the reconstructions of the autoencoder. For comparision, we use an injective embedding[16] of a 64-dimensional hyperplane in $X$. The latter is equivalent to sampling 784-dimensional vectors from $U([0, 1])$ and setting all but the first 64 components to zero. The target function is white-noise, sampled as scalars from the uniform distribution $U([0, 1])$. Two identical networks are trained under identical conditions, and Fig 13 shows the resulting loss curves, each averaged over 10 runs.

This result complements the findings of Arpit et al. (2017) and Zhang et al. (2017a), which show that it's easier to fit random labels to random inputs if the latter is defined on the full dimensional input space (i.e. the dimension of the flat manifold is the same as that of the input space, and not that of the underlying data-manifold being used for comparison).

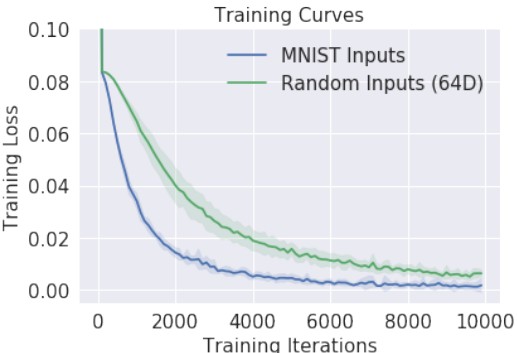

Figure 13: Loss curves of two identical networks trained to regress white-noise under identical conditions, one on MNIST reconstructions from a DAE with 64 encoder features (blue), and the other on 64-dimensional random vectors (green).

## A.5 CIFAR-10: IT'S ALL CONNECTED

We have seen that deep neural networks are biased towards learning low frequency functions. This should have as a consequence that isolated *bubbles* of constant prediction are rare. This in turn implies that given any two points in the input space and a network function that predicts the same class for the said points, there should be a path connecting them such that the network prediction does not change along the path. In the following, we present an experiment where we use a path finding method to find such a path between all Cifar-10 input samples indeed exist.

**Experiment 7.** Using AutoNEB Kolsbjerg et al. (2016), we construct paths between (adversarial) Cifar-10 images that are classified by a ResNet20 to be all of the same target class. AutoNEB bends

---

[15]This experiment yields the same result if variational autoencoders are used instead.
[16]The xy-plane is $\mathbb{R}^3$ an injective embedding of a subset of $\mathbb{R}^2$ in $\mathbb{R}^3$.

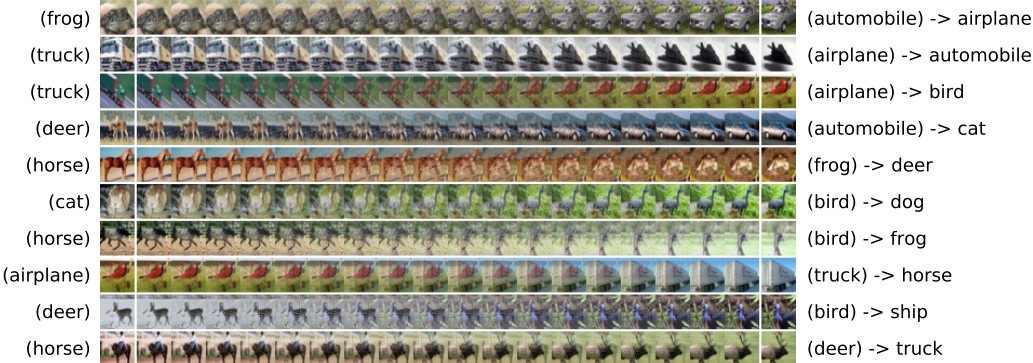

Figure 14: Path between CIFAR-10 adversarial examples (e.g. "frog" and "automobile", such that all images are classified as "airplane").

a linear path between points in some space $\mathbb{R}^m$ so that some maximum energy along the path is minimal. Here, the space is the input space of the neural network, i.e. the space of $32 \times 32 \times 3$ images and the logit output of the ResNet20 for a given class is minimized. We construct paths between the following points in image space:

- From one training image to another,

- from a training image to an adversarial,

- from one adversarial to another.

We only consider pairs of images that belong to the same class $c$ (or, for adversarials, that originate from another class $\neq c$, but that the model classifies to be of the specified class $c$). For each class, we randomly select 50 training images and select a total of 50 random images from all other classes and generate adversarial samples from the latter. Then, paths between all pairs from the whole set of images are computed.

The AutoNEB parameters are chosen as follows: We run four NEB iterations with 10 steps of SGD with learning rate 0.001 and momentum 0.9. This computational budget is similar to that required to compute the adversarial samples. The gradient for each NEB step is computed to maximize the logit output of the ResNet-20 for the specified target class $c$. We use the formulation of NEB without springs Draxler et al. (2018).

The result is very clear: We can find paths between *all* pairs of images for all CIFAR10 labels that do not cross a single decision boundary. This means that all paths belong to the same connected component regarding the output of the DNN. This holds for all possible combinations of images in the above list. Figure 15 shows connecting training to adversarial images and Figure 14 paths between pairs of adversarial images. Paths between training images are not shown, they provide no further insight. Note that the paths are strikingly simple: Visually, they are hard to distinguish from the linear interpolation. Quantitatively, they are essentially (but not exactly) linear, with an average length $(3.0 \pm 0.3)\%$ longer than the linear connection.

## B    BRIEF RECAPITULATION OF FOURIER ANALYSIS

The Fourier transform is a powerful mathematical tool used to represent functions as a weighted sum of oscillating functions, given that the function satisfies certain conditions. In the realm of signal processing and beyond, it is used to represent a time (space) domain signal $f$ as a sum of sinusoids of various (spatial) frequencies $\mathbf{k}$, where the weights are referred to as the Fourier coefficients $\tilde{f}(\mathbf{k})$.

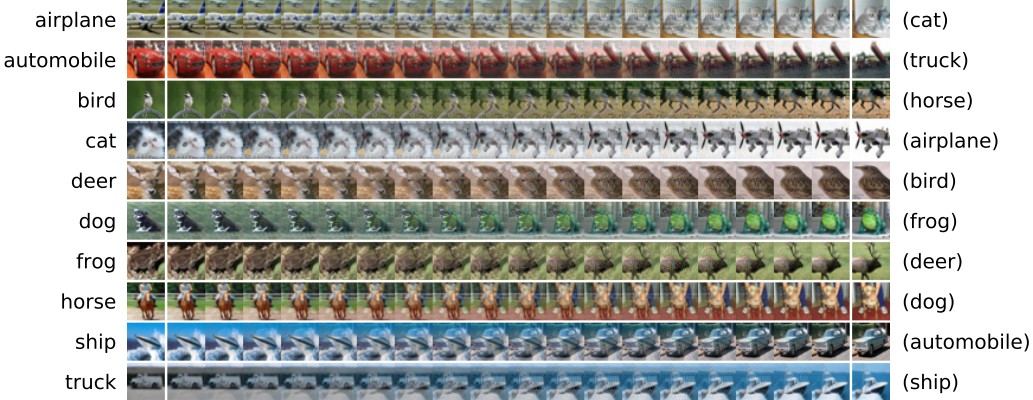

| | | |
|---|---|---|
| airplane | | (cat) |
| automobile | | (truck) |
| bird | | (horse) |
| cat | | (airplane) |
| deer | | (bird) |
| dog | | (frog) |
| frog | | (deer) |
| horse | | (dog) |
| ship | | (automobile) |
| truck | | (ship) |

Figure 15: Each row is a path through the image space from an adversarial sample (right) to a true training image (left). All images are classified by a ResNet-20 to be of the class of the training sample on the right with at least 95% softmax certainty. This experiment shows we can find a path from adversarial examples (right, Eg. "(cat)") that are classified as a particular class ("airplane") are connected to actual training samples from that class (left, "airplane") such that all samples along that path are also predicted by the network to be of the same class.

Let $f : \mathbb{R}^n \rightarrow \mathbb{R}$ be a squared-integrable function[17], i.e. such that $\int_{\mathbf{x} \in \mathbb{R}^n} |f(\mathbf{x})|^2 d\mathbf{x}$ is finite, or $f \in L^2(\mathbb{R}^n)$. With the Fourier inversion theorem, it holds:

$$f(\mathbf{x}) = \frac{1}{2\pi} \int_{\mathbf{k} \in \mathbb{R}^n} \tilde{f}(\mathbf{k}) e^{i\mathbf{k} \cdot \mathbf{x}} d\mathbf{k} \qquad (17) \qquad \tilde{f}(\mathbf{k}) = \int_{\mathbf{x} \in \mathbb{R}^n} f(\mathbf{x}) e^{-i\mathbf{k} \cdot \mathbf{x}} d\mathbf{x} \qquad (18)$$

Informally, equation 17 expresses the function $f(\mathbf{x})$ as a weighted sum (the integral) of plane waves $e^{\pm i\mathbf{k} \cdot \mathbf{x}}$ of the angular wavenumber $\mathbf{k}$, where the unit vector $\hat{\mathbf{k}}$ gives the direction of the corresponding wave in $n$-D space and the magnitude $k$ is inversely proportional to the wavelength. Equation 18 gives the expression for $\tilde{f}(\mathbf{k})$, which is called the Fourier transform or the *Fourier spectrum* or simply the spectrum of $f$. The $\frac{1}{2\pi}$ coefficient and sign in the exponential functions are matters of convention.

The asymptotic behaviour of $\tilde{f}$ for $k \rightarrow \infty$ is a measure of smoothness of $f$. In Bachmann-Landau or *asymptotic* notation[18], we say $\tilde{f} = \mathcal{O}(k^{-1})$ if for $k \rightarrow \infty$, the function $\tilde{f}$ decays at least as fast as $\frac{1}{k}$. A function whose spectrum is $\mathcal{O}(k^{-2})$ is in a sense smoother than one whose spectrum is $\mathcal{O}(k^{-1})$, while the spectrum of an infinitely differentiable (or *smooth*) function must decay faster than any rational function of $k$, assuming the function is integrable, i.e. the integral of its absolute value over its domain is finite (or the function is $L^1$). Intuitively, the higher-frequency oscillations in a smoother function must vanish faster. Formally, this is a straightforward consequence of the Riemann-Lebesgue lemma, stating that the spectrum of any $L^1$ function must vanish at infinity (potentially arbitrarily slowly), taken together with the well known property of the Fourier transform that it diagonalizes the differential operator i.e. $[\widetilde{\nabla_{\mathbf{x}} f}](\mathbf{k}) = \mathbf{k} \tilde{f}(\mathbf{k})$.

---

[17]On a formal note, the squared integrability is only required for the inverse Fourier transform to exist; for the forward transform, integrability is enough. Moreover, the Fourier transform can be generalized to tempered distributions, which allow for evaluating the Fourier coefficients of non-integrable e.g. non-zero constant functions.

[18]Formally, $f = \mathcal{O}(g) \implies \limsup_{x \rightarrow \infty} \left| \frac{f(x)}{g(x)} \right| < \infty$.

## C  THE CONTINUOUS PIECEWISE LINEAR STRUCTURE OF DEEP ReLU NETWORKS

We consider the class of ReLU network functions $f : \mathbb{R}^d \mapsto \mathbb{R}$ defined by Eqn. 1. Following the terminology of Raghu et al. (2016); Montufar et al. (2014), each linear region of the network then corresponds to a unique *activation pattern*, wherein each hidden neuron is assigned an activation variable $\epsilon \in \{-1, 1\}$, conditioned on whether its input is positive or negative. ReLU networks can be explictly expressed as a sum over all possible activation patterns, as in the following lemma.

**Lemma 3.** *Given $L$ binary vectors $\epsilon^{(1)}, \cdots \epsilon^{(L)}$ with $\epsilon^{(k)} \in \{-1, 1\}^{d_k}$, let $T^{(k)}_{\epsilon^{(k)}} : \mathbb{R}^{d_{k-1}} \to \mathbb{R}^{d_k}$ the affine function defined by $T^{(k)}_{\epsilon^{(k)}}(\mathbf{u})_i = (T^{(k)}(\mathbf{u}))_i$ if $(\epsilon_k)_i = 1$, and $0$ otherwise. ReLU network functions, as defined in Eqn. 1, can be expressed as*

$$f(\mathbf{x}) = \sum_{\epsilon^{(1)}, \cdots \epsilon^{(L)}} 1_{P_{f,\epsilon}}(\mathbf{x}) \left( T^{(L+1)} \circ T^{(L)}_{\epsilon^{(L)}} \circ \cdots \circ T^{(1)}_{\epsilon^{(1)}} \right)(\mathbf{x}) \tag{19}$$

*where $1_P$ denotes the indicator function of the subset $P \subset \mathbb{R}^d$, and $P_{f,\epsilon}$ is the polytope defined as the set of solutions of the following linear inequalities (for all $k = 1, \cdots, L$):*

$$(\epsilon_k)_i \, (T^{(k)} \circ T^{(k-1)}_{\epsilon^{(k-1)}} \circ \cdots \circ T^{(1)}_{\epsilon^{(1)}})(\mathbf{x})_i \geq 0, \quad i = 1, \cdots d_k \tag{20}$$

$f$ is therefore affine on each of the polytopes $P_{f,\epsilon}$, which finitely partition the input space $\mathbb{R}^d$ to convex polytopes. Remarkably, the correspondence between ReLU networks and CPWL functions goes both ways: Arora et al. (2018) show that every CPWL function is be represented by a ReLU network, which in turn endows ReLU networks with the universal approximation property.

Finally, in the standard basis, each affine map $T^{(k)} : \mathbb{R}^{d_{k-1}} \to \mathbb{R}^{d_k}$ is specified by a weight matrix $W^{(k)} \in \mathbb{R}^{d_{k-1}} \times \mathbb{R}^{d_k}$ and a bias vector $b^{(k)} \in \mathbb{R}^{d_k}$. In the linear region $P_{f,\epsilon}$, $f$ can be expressed as $f_\epsilon(x) = W_\epsilon x + b_\epsilon$, where in particular

$$W_\epsilon = W^{(L+1)} W^{(L)}_{\epsilon_L} \cdots W^{(1)}_{\epsilon_1} \in \mathbb{R}^{1 \times d}, \tag{21}$$

where $W^{(k)}_\epsilon$ is obtained from $W^{(k)}$ by setting its $j$th column to zero whenever $(\epsilon_k)_j = -1$.

## D  FOURIER ANALYSIS OF ReLU NETWORKS

### D.1  PROOF OF LEMMA 1

*Proof.* The vector-valued function $\mathbf{k} f(\mathbf{x}) e^{i\mathbf{k} \cdot \mathbf{x}}$ is continuous everywhere and has well-defined and continuous gradients almost everywhere. So by Stokes' theorem (see e.g Spivak (2018)), the integral of its divergence is a pure boundary term. Since we restricted to functions with compact support, the theorem yields

$$\int \nabla_\mathbf{x} \cdot \left[ \mathbf{k} f(\mathbf{x}) e^{-i\mathbf{k} \cdot \mathbf{x}} \right] d\mathbf{x} = 0 \tag{22}$$

The integrand is $(\mathbf{k} \cdot (\nabla_\mathbf{x} f)(\mathbf{x}) - ik^2 f(\mathbf{x})) e^{-i\mathbf{k} \cdot \mathbf{x}}$, so we deduce,

$$\hat{f}(\mathbf{k}) = \frac{1}{-ik^2} \mathbf{k} \cdot \int (\nabla_\mathbf{x} f)(\mathbf{x}) \, e^{-i\mathbf{k} \cdot \mathbf{x}} \tag{23}$$

Now, within each polytope of the decomposition (19), $f$ is affine so its gradient is a constant vector, $\nabla_\mathbf{x} f_\epsilon = W_\epsilon^T$, which gives the desired result (1). □

### D.2  FOURIER TRANSFORM OF POLYTOPES

#### D.2.1  THEOREM 1 OF DIAZ ET AL. (2016)

Let $F$ be a $m$ dimensional polytope in $\mathbb{R}^d$, such that $1 \leq m \leq d$. Denote by $\mathbf{k} \in \mathbb{R}^d$ a vector in the Fourier space, by $\phi_\mathbf{k}(x) = -\mathbf{k} \cdot \mathbf{x}$ the linear phase function, by $\tilde{F}$ the Fourier transform of the indicator function on $F$, by $\partial F$ the boundary of $F$ and by $\text{vol}_m$ the $m$-dimensional (Hausdorff) measure. Let $\text{Proj}_F(\mathbf{k})$ be the orthogonal projection of $\mathbf{k}$ on to $F$ (obtained by removing all components of $\mathbf{k}$ orthogonal to $F$). Given a $m-1$ dimensional facet $G$ of $F$, let $\mathbf{N}_F(G)$ be the unit normal vector to $G$ that points out of $F$. It then holds:

1. If $\text{Proj}_F(\mathbf{k}) = 0$, then $\phi_\mathbf{k}(x) = \Phi_\mathbf{k}$ is constant on $F$, and we have:

$$\tilde{F} = \text{vol}_F(F) e^{i\Phi_\mathbf{k}} \tag{24}$$

2. But if $\text{Proj}_F(\mathbf{k}) \neq 0$, then:

$$\tilde{F} = i \sum_{G \in \partial F} \frac{\text{Proj}_F(\mathbf{k}) \cdot \mathbf{N}_F(G)}{\|\text{Proj}_F(\mathbf{k})\|^2} \tilde{G}(\mathbf{k}) \tag{25}$$

### D.2.2  DISCUSSION

The above theorem provides a recursive relation for computing the Fourier transform of an arbitrary polytope. More precisely, the Fourier transform of a $m$-dimensional polytope is expressed as a sum of fourier transforms over the $m-1$ dimensional boundaries of the said polytope (which are themselves polytopes) times a $\mathcal{O}(k^{-1})$ *weight* term (with $k = \|\mathbf{k}\|$). The recursion terminates if $\text{Proj}_F(\mathbf{k}) = 0$, which then yields a constant.

To structure this computation, Diaz et al. (2016) introduce a book-keeping device called the *face poset* of the polytope. It can be understood as a weighted tree diagram with polytopes of various dimensions as its nodes. We start at the root node which is the full dimensional polytope $P$ (i.e. we initially set $m = n$). For all of the codimension-one boundary faces $F$ of $P$, we then draw an edge from the root $P$ to node $F$ and weight it with a term given by:

$$W_{F,G} = i \frac{\text{Proj}_F(\mathbf{k}) \cdot \mathbf{N}_F(G)}{\|\text{Proj}_F(\mathbf{k})\|^2} \tilde{G}(\mathbf{k}) \tag{26}$$

and repeat the process iteratively for each $F$. Note that the weight term is $\mathcal{O}(k^{-1})$ where $\text{Proj}_F(\mathbf{k}) \neq 0$. This process yields tree paths $T : P \to F_1 \to ... \to F_q$ where each $F_{i+1} \in \partial F_i$ has one dimension less than $F_i$. For a given path and $\mathbf{k}$, the terminal node for this path, $F_q$, is the first polytope for which $\text{Proj}_{F_q}(\mathbf{k}) = 0$. The final Fourier transform is obtained by multiplying the weights along each path and summing over all tree paths:

$$\tilde{1}_P(\mathbf{k}) = \sum_T i^q \prod_{i=0}^{q-1} \frac{\text{Proj}_{F_i}(\mathbf{k}) \cdot \mathbf{N}_{F_i}(F_{i+1})}{\|\text{Proj}_{F_i}(\mathbf{k})\|^2} \text{vol}_{F_q}(F_q) e^{i\Phi_\mathbf{k}} \tag{27}$$

where we wrote $F_0 = P$. Together with Lemma 1, this gives the closed form expression of the Fourier transform of ReLU networks.

For a generic vector $\mathbf{k}$, all paths terminate at the zero-dimensional vertices of the original polytope, i.e. $\dim(F_q) = 0$, implying the length of the path $q$ equals the number of dimensions $d$, yielding a $\mathcal{O}(k^{-d})$ spectrum. The exceptions occur if a path terminates prematurely, because $\mathbf{k}$ happens to lie orthogonal to some $d - r$-dimensional face $F_r$ in the path, in which case we are left with a $\mathcal{O}(k^{-r})$ term (with $r < d$) which dominates asymptotically. Note that all vectors orthogonal to the $d - r$ dimensional face $F_r$ lie on a $r$-dimensional subspace of $\mathbb{R}^d$. Since a polytope has a finite number of faces (of any dimension), the $\mathbf{k}$'s for which the Fourier transform is $\mathcal{O}(k^{-r})$ (instead of $\mathcal{O}(k^{-d})$) lies on a finite union of closed subspaces of dimension $r$ (with $r < d$). The Lebesgue measure of all such lower dimensional subspaces for all such $r$ is 0, leading us to the conclusion that the spectrum decays as $\mathcal{O}(k^{-d})$ for *almost all* $\mathbf{k}$'s. We formalize this in the following corollary.

**Corollary 1.** *Let $P$ be a full dimensional polytope in $\mathbb{R}^n$. The Fourier spectrum of its indicator function $\tilde{1}_P$ satisfies the following:*

$$|\tilde{1}_P(\mathbf{k})| = \mathcal{O}\left(\frac{1}{k^{\Delta_\mathbf{k}}}\right) \tag{28}$$

*where $1 \leq \Delta_\mathbf{k} \leq n$, and $\Delta_\mathbf{k} = j$ for $\mathbf{k}$ on a finite union of $j$-dimensional subspaces of $\mathbb{R}^n$.*

### D.3  PROOF OF THE LIPSCHTIZ BOUND

**Proposition 2.** *The Lipschitz constant $L_f$ of the ReLU network $f$ is bound as follows (for all $\epsilon$):*

$$\|W_\epsilon\| \leq L_f \leq \prod_{k=1}^{L+1} \|W^{(k)}\| \leq \|\theta\|_\infty^{L+1} \sqrt{d} \prod_{k=1}^{L} d_k \tag{29}$$

*Proof.* The first equality is simply the fact that $L_f = \max_\epsilon \|W_\epsilon\|$, and the second inequality follows trivially from the parameterization of a ReLU network as a chain of function compositions[19], together with the fact that the Lipschitz constant of the ReLU function is 1 (cf. Miyato et al. (2018), equation 7). To see the third inequality, consider the definition of the spectral norm of a $I \times J$ matrix $W$:

$$\|W\| = \max_{\|\mathbf{h}\|=1} \|W\mathbf{h}\| \tag{30}$$

Now, $\|W\mathbf{h}\| = \sqrt{\sum_i |\mathbf{w}_i \cdot \mathbf{h}|}$, where $\mathbf{w}_i$ is the $i$-th row of the weight matrix $W$ and $i = 1, ..., I$. Further, if $\|\mathbf{h}\| = 1$, we have $|\mathbf{w}_i \cdot \mathbf{h}| \leq \|\mathbf{w}_i\|\|\mathbf{h}\| = \|\mathbf{w}_i\|$. Since $\|\mathbf{w}_i\| = \sqrt{\sum_j |w_{ij}|}$ (with $j = 1, ..., J$) and $|w_{ij}| \leq \|\theta\|_\infty$, we find that $\|\mathbf{w}_i\| \leq \sqrt{J}\|\theta\|_\infty$. Consequently, $\sqrt{\sum_i |\mathbf{w}_i \cdot \mathbf{h}|} \leq \sqrt{IJ}\|\theta\|_\infty$ and we obtain:

$$\|W\| \leq \sqrt{IJ}\|\theta\|_\infty \tag{31}$$

Now for $W = W^{(k)}$, we have $I = d_{k-1}$ and $J = d_k$. In the product over $k$, every $d_k$ except the first and the last occur in pairs, which cancels the square root. For $k = 1$, $d_{k-1} = d$ (for the $d$ input neurons) and for $k = L + 1$, $d_k = 1$ (for a single output neuron). The final inequality now follows. $\qquad\square$

### D.4 THE FOURIER TRANSFORM OF A FUNCTION COMPOSITION

Consider Equation 14. The general idea is to investigate the behaviour of $P_\gamma(\mathbf{l}, \mathbf{k})$ for large frequencies $\mathbf{l}$ on manifold but smaller frequencies $\mathbf{k}$ in the input domain. In particular, we are interested in the regime where the stationary phase approximation is applicable to $P_\gamma$, i.e. when $l^2 + k^2 \to \infty$ (cf. section 3.2. of Bergner et al.). In this regime, the integrand in $P_\gamma(\mathbf{k}, \mathbf{l})$ oscillates fast enough such that the only constructive contribution originates from where the phase term $u(\mathbf{z}) = \mathbf{k} \cdot \gamma(\mathbf{z}) - \mathbf{l} \cdot \mathbf{z}$ does not change with changing $\mathbf{z}$. This yields the condition that $\nabla_\mathbf{z} u(\mathbf{z}) = 0$, which translates to the condition (with Einstein summation convention implied and $\partial_\nu = \partial/\partial x_\nu$):

$$l_\nu = k_\mu \partial_\nu \gamma_\mu(\mathbf{z}) \tag{32}$$

Now, we impose periodic boundary conditions[20] on the components of $\gamma$, and without loss of generality we let the period be $2\pi$. Further, we require that the manifold be contained in a box[21] of some size in $\mathbb{R}^d$. The $\mu$-th component $\gamma_\mu$ can now be expressed as a Fourier series:

$$\gamma_\mu(\mathbf{z}) = \sum_{\mathbf{p} \in \mathbb{Z}^m} \tilde{\gamma}_\mu[\mathbf{p}]e^{-ip_\rho z_\rho} \tag{33} \qquad \partial_\nu \gamma_\mu(\mathbf{z}) = \sum_{\mathbf{p} \in \mathbb{Z}^m} -ip_\nu \tilde{\gamma}_\mu[\mathbf{p}]e^{-ip_\rho z_\rho} \tag{34}$$

Equation 34 can be substituted in equation 32 to obtain:

$$l\hat{l}_\nu = -ik \sum_{\mathbf{p} \in \mathbb{Z}^m} p_\nu \hat{k}_\mu \tilde{\gamma}_\mu[\mathbf{p}]e^{-ip_\rho z_\rho} \tag{35}$$

where we have split $k_\mu$ and $l_\nu$ in to their magnitudes $k$ and $l$ and directions $\hat{k}_\nu$ and $\hat{l}_\mu$ (respectively). We are now interested in the conditions on $\gamma$ under which the RHS can be large in magnitude, even when $k$ is fixed. Recall that $\gamma$ is constrained to a box – consequently, we can not arbitrarily scale up $\tilde{\gamma}_\mu$. However, if $\tilde{\gamma}_\mu[\mathbf{p}]$ decays slowly enough with increasing $\mathbf{p}$, the RHS can be made arbitrarily large (for certain conditions on $\mathbf{z}$, $\hat{l}_\mu$ and $\hat{k}_\nu$).

### E VOLUME IN PARAMETER SPACE AND PROOF OF PROPOSITION 1

For a given neural network, we now show that the volume of the parameter space containing parameters that contribute $\epsilon$-non-negligibly to frequency components of magnitude $k'$ above a certain

---

[19]Recall that the Lipschitz constant of a composition of two or more functions is the product of their respective Lipschtiz constants.

[20]This is possible whenever $\gamma$ is defined on a bounded domain, e.g. on $[0, 1]^m$.

[21]This is equivalent to assuming that the data lies in a bounded set.

cut-off $k$ decays with increasing $k$. For notational simplicity and without loss of generality, we absorb the direction $\hat{\mathbf{k}}$ of $\mathbf{k}$ in the respective mappings and only deal with the magnitude $k$.

**Definition 1.** *Given a ReLU network $f_\theta$ of fixed depth, width and weight clip $K$ with parameter vector $\theta$, an $\epsilon > 0$ and $\Theta = B_K^\infty(0)$ a $L^\infty$ ball around 0, we define:*

$$\Xi_\epsilon(k) = \{\theta \in \Theta | \exists k' > k, |\tilde{f}_\theta(k')| > \epsilon\}$$

*as the set of all parameters vectors $\theta \in \Xi_\epsilon(k)$ that contribute more than an $\epsilon$ in expressing one or more frequencies $k'$ above a cut-off frequency $k$.*

**Remark 1.** *If $k_2 \geq k_1$, we have $\Xi_\epsilon(k_2) \subseteq \Xi_\epsilon(k_1)$ and consequently $vol(\Xi_\epsilon(k_2)) \leq vol(\Xi_\epsilon(k_1))$, where vol is the Lebesgue measure.*

**Lemma 4.** *Let $1_k^\epsilon(\theta)$ be the indicator function on $\Xi_\epsilon(k)$. Then:*

$$\exists \kappa > 0 : \forall k \geq \kappa, 1_k^\epsilon(\theta) = 0$$

*Proof.* From theorem 1, we know that[22] $|\tilde{f}_\theta(k)| = \mathcal{O}(k^{-\Delta-1})$ for an integer $1 \leq \Delta \leq d$. In the worse case where $\Delta = 1$, we have that $\exists M < \infty : |\tilde{f}_\theta(k)| < \frac{M}{k^2}$. Now, simply select a $\kappa > \sqrt{\frac{M}{\epsilon}}$ such that $\frac{M}{\kappa^2} < \epsilon$. This yields that $|\tilde{f}_\theta(\kappa)| < \frac{M}{\kappa^2} < \epsilon$, and given that $\frac{M}{\kappa^2} \leq \frac{M}{k^2} \forall k \geq \kappa$, we find $|\tilde{f}_\theta(k)| < \epsilon \forall k \geq \kappa$. Now by definition 1, $\theta \notin \Xi_\epsilon(\kappa)$, and since $\Xi_\epsilon(k) \subseteq \Xi_\epsilon(\kappa)$ (see remark 1), we have $\theta \notin \Xi_\epsilon(k)$, implying $1_k^\epsilon(\theta) = 0 \forall k \geq \kappa$. $\square$

**Remark 2.** *We have $1_k^\epsilon(\theta) \leq |\tilde{f}_\theta(k)|$ for large enough $k$ (i.e. for $k \geq \kappa$), since $|\tilde{f}_\theta(k)| \geq 0$.*

**Proposition 1.** *The relative volume of $\Xi_\epsilon(k)$ w.r.t. $\Theta$ is $\mathcal{O}(k^{-\Delta-1})$ where $1 \leq \Delta \leq d$.*

*Proof.* The volume is given by the integral over the indicator function, i.e.

$$\text{vol}(\Xi_\epsilon(k)) = \int_{\theta \in \Theta} 1_k^\epsilon(\theta) d\theta$$

For a large enough $k$, we have from remark 2, the monotonicity of the Lebesgue integral and theorem 1 that:

$$\text{vol}(\Xi_\epsilon(k)) = \int_{\theta \in \Theta} 1_k^\epsilon(\theta) d\theta \leq \int_{\theta \in \Theta} |\tilde{f}_\theta(k)| d\theta = \mathcal{O}(k^{-d+\Delta-1}) \text{vol}(\Theta)$$

$$\implies \frac{\text{vol}(\Xi_\epsilon(k))}{\text{vol}(\Theta)} = \mathcal{O}(k^{-\Delta-1})$$

$\square$

# F  KERNEL MACHINES AND KNNS

In this section, in light of our findings, we want to compare DNNs with K-nearest neighbor (k-NN) classifier and kernel machines which are also popular learning algorithms, but are, in contrast to DNNs, better understood theoretically.

## F.1  KERNEL MACHINES VS DNNS

Given that we study why DNNs are biased towards learning smooth functions, we note that kernel machines (KM) are also highly Lipschitz smooth (Eg. for Gaussian kernels all derivatives are bounded). However there are crutial differences between the two. While kernel machines can approximate any target function in principal (Hammer & Gersmann, 2003), the number of Gaussian kernels needed scales linearly with the number of sign changes in the target function (Bengio et al., 2009). Ma & Belkin (2017) have further shown that for smooth kernels, a target function cannot be approximated within $\epsilon$ precision in any polynomial of $1/\epsilon$ steps by gradient descent.

---

[22]Note that in theorem 1, $\Delta_{\hat{\mathbf{k}}}$ depends only on the direction of $\mathbf{k}$, which we absorb in the definition of $\Delta$.

Deep networks on the other hand are also capable of approximating any target function (as shown by the universal approximation theorems Hornik et al. (1989); Cybenko (1989)), but they are also parameter efficient in contrast to KM. For instance, we have seen that deep ReLU networks separate the input space into number of linear regions that grow polynomially in width of layers and exponentially in the depth of the network (Montufar et al., 2014; Raghu et al., 2016). A similar result on the exponentially growing expressive power of networks in terms of their depth is also shown in (Poole et al., 2016). In this paper we have further shown that DNNs are inherently biased towards lower frequency (smooth) functions over a finite parameter space. This suggests that DNNs strike a good balance between function smoothness and expressibility/parameter-efficiency compared with KM.

## F.2  K-NN CLASSIFIER VS. DNN CLASSIFIER

$K$-nearest neighbor ($K$NN) also has a historical importance as a classification algorithm due to its simplicity. It has been shown to be a consistent approximator Devroye et al. (1996), i.e., asymptotically its empirical risk goes to zero as $K \to \infty$ and $K/N \to 0$, where $N$ is the number of training samples. However, because it is a memory based algorithm, it is prohibitively slow for large datasets. Since the smoothness of a $K$NN prediction function is not well studied, we compare the smoothness between $K$NN and DNN. For various values of $K$, we train a $K$NN classifier on a $k = 150$ frequency signal (which is binarized) defined on the $L = 20$ manifold (see section 4), and extract probability predictions on a box interval in $\mathbb{R}^2$. On this interval, we evaluate the 2D FFT and integrate out the angular components to obtain $\zeta(k)$:

$$\zeta(k) = \frac{d}{dk} \int_0^k dk'k' \int_0^{2\pi} d\varphi |\tilde{f}(k', \varphi)| \tag{36}$$

Finally, we plot $\zeta(k)$ for various $K$ in figure 16e. Furthermore, we train a DNN on the very same dataset and overlay the radial spectrum of the resulting probability map on the same plot. We find that while DNN's are as expressive as a $K = 1$ KNN classifier at lower (radial) frequencies, the frequency spectrum of DNNs decay faster than KNN classifier for all values of $K$ considered, indicating that the DNN is smoother than the $K$NNs considered. We also repeat the experiment corresponding to Fig. 4 with KNNs (see Fig. 16) for various $K$'s, to find that unlike DNNs, KNNs do not necessarily perform better for larger $L$'s, suggesting that KNNs do not exploit the geometry of the manifold like DNNs do.

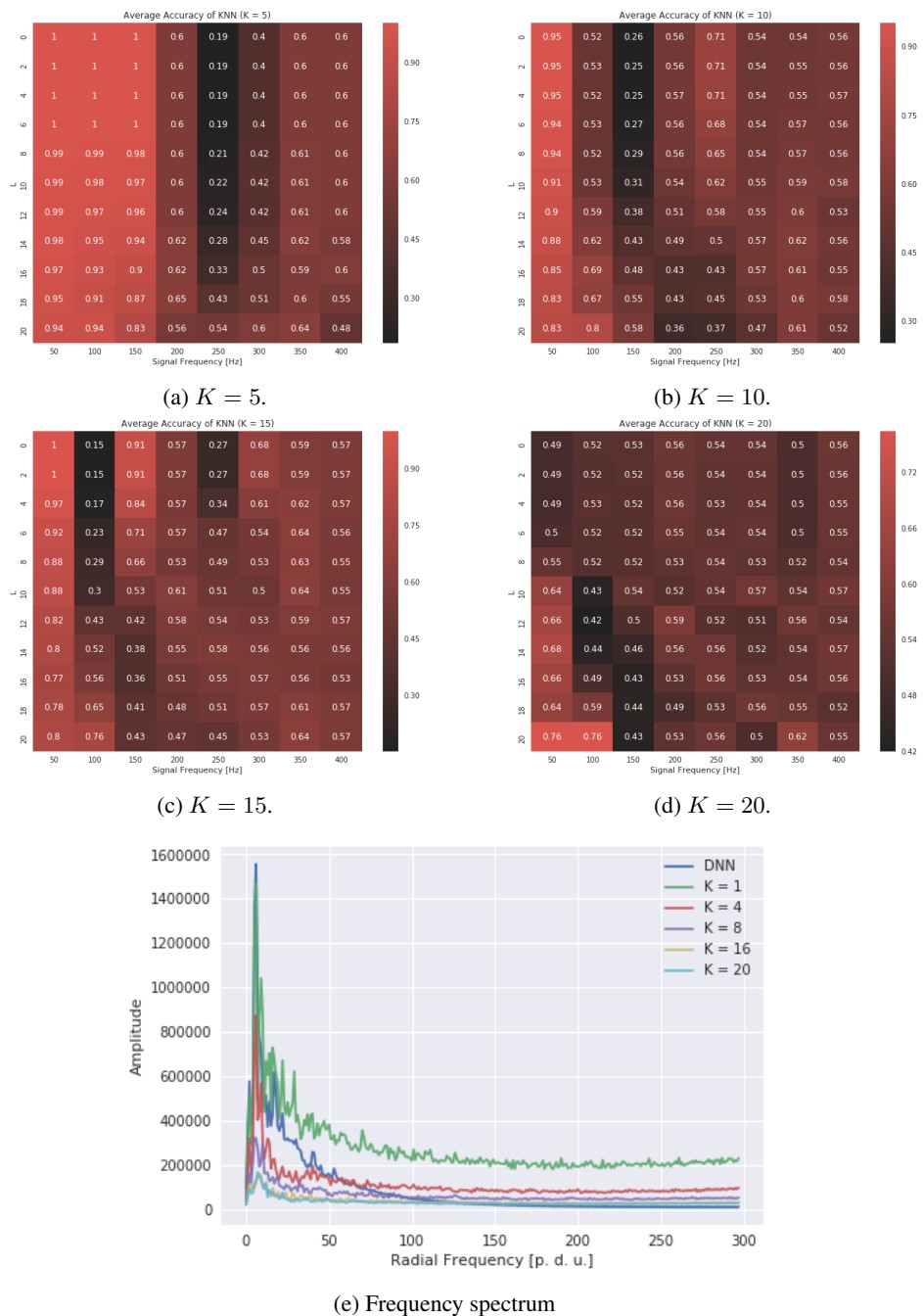

(e) Frequency spectrum

Figure 16: (a,b,c,d): Heatmaps of training accuracies ($L$-vs-$k$) of KNNs for various $K$. When comparing with figure 4, note that the y-axis is flipped. (e): The frequency spectrum of $K$NNs with different values of $K$, and a DNN. The DNN learns a smoother function compared with the $K$NNs considered since the spectrum of the DNN decays faster compared with $K$NNs.

