# OpenReview forum: "On the Spectral Bias of Neural Networks"
_ICLR.cc/2019/Conference_

### Official Review · AnonReviewer1 · 2018-11-02
**Analysis of Spectral Bias of ReLU networks**

**Rating:** 6
**Confidence:** 3

**Review:**

Analysis of Spectral Bias of ReLU networks

The paper uses Fourier analysis to study ReLU network utilizing its continuous piecewise linear structure.

Main finding is that these networks are biased towards learning low frequency which authors denote `spectral bias’.  This provides another theoretical perspective of neural networks preferring more smooth functions while being able to fit complicated function. Also shows that in terms of parameters networks representing lower frequency modes are more robust.

Pro:
- Nice introduction to Fourier analysis providing non-trivial insights of ReLU networks.
- Intuitive toy experiments to show spectral bias and its properties
- Thorough theoretical analysis and empirical support

Con:
- The analysis is clearly for ReLU networks although the title may provide a false impression that it corresponds to general networks with other non-linearities. It is an interesting question whether the behaviour characterized by the authors are universal.
- At least for me, Section 4 was not as clearly presented as other section. It takes more effort to parse what experiments were conducted and why such experiments are provided.
- Although some experiments on real dataset are provided in the appendix, I personally could not read much intuition of theoretical findings to the networks used in practice. Does the spectral bias suggest better way of training or designing neural networks for example?

Comments/Questions:
- In Figure 1, two experiments show different layerwise behaviour, i.e. equal amplitude experiment (a) shows spectral norm evolution for all the layers are almost identical whereas in increasing amplitude experiment (b) shows higher layer change spectral norm more than the lower layer. Do you understand why and does Fourier spectrum provide insights into layerwise behaviour?
- Experiment 3 seems to perform binary classification using thresholding to the logits. But how do you find these results also hold for cross-entropy loss?
“The results confirm the behaviour observed in Experiment 2, but in the case of classification tasks with categorical cross-entropy loss.”


Nit: p3 ReLu -> ReLU / p5 k \in {50, 100, … 350, 400} (close bracket) / p5 in Experiment 2 and 3 descriptions the order of Figure appears flipped. Easier to read if the figure appears as the paper reads / p7 Equation 11 [0, 1]^m


********* updated review *************

Based on the issues raised from other reviewers and rebuttal from authors, I started to share some of the concerns on applicability of Thm 1 in obtaining information on low k Fourier coefficients. Although I empathize author's choice to mainly analyze synthetic data, I think it is critical to show the decays for moderately large k in realistic datasets. It will convince other reviewers of significance of main result of the paper.

---

> ### Author Response · Authors · 2018-11-12
> **Response to Reviewer 1**
>
> Thank you for your insightful comments and thought provoking questions!
>
> **On the choice of activation**:
>
> > It is an interesting question whether the behaviour characterized by the authors are universal.
>
> Indeed! We suspect it is fairly common for the usual activation functions. For instance, Xu et al (2018) (in an independent work made available online shortly after ours) show the same behaviour for deep networks with sigmoid/tanh activation.
>
> **On clarify of Section 4**:
>
> Thanks for this valuable feedback. Section 4 has been streamlined. We hope it's nicer to read now!
>
> **On the practical significance of the results**:
>
> We believe that the potent toolbox of Fourier analysis has a lot of unharnessed potential for shedding light on fundamental issues like generalization (e.g. from a sampling theory perspective - under what conditions can the target function be reconstructed from samples?) and adversarial examples (e.g. from the perspective of sensitivity analysis under data-distributions supported on low-dimensional manifolds). We view the spectral bias as an important Fourier domain consequence of the prior of neural net parameterization, and as such, expect it to play an important role in future work.
>
> **About your comments/questions**:
>
> > Do you understand why and does Fourier spectrum provide insights into layerwise behaviour?
>
> We consider it an exciting open question! We tried to look for a pattern in the layer-wise behaviour of the spectral norms, but did not find anything statistical significant over multiple runs (except that they were consistently increasing, see e.g. this plot: https://imgur.com/a/rCbAW47 ).
>
> > how do you find these results also hold for cross-entropy loss?
>
> We might not have been clear enough on the set up of Experiment 3: in fact we do use cross-entropy loss there. We threshold a sinusoid at 0.5, and train a network on the resulting binary target signal using binary cross-entropy loss (we use Pytorch's BCEWithLogitsLoss, which takes a sigmoid internally). The "categorical" is indeed a typo, it should be "binary". We have rephrased the corresponding lines in the latest revision and hope it's clearer now. We have also fixed the typos you found (thank you!).
>
> In closing:
>
> Thank you for the positive feedback. We hope to have adequately addressed your concerns.
>
> [Xu et al 2018] https://arxiv.org/abs/1807.01251

---

> > ### Comment · AnonReviewer1 · 2018-12-03
> > **Thank you for the response**
> >
> > I thank the authors for taking time and effort to address the issues raised.
> >
> > Personally the synthetic toy data set experiments are informative and I agree with authors that since other work show some hints of universality of the spectral bias, there's utility of studying controlled settings. Therefore I am not concerned with experimental settings as other reviewers are.

---

### Official Review · AnonReviewer2 · 2018-11-04
**interesting ideas; message unclear**

**Rating:** 5
**Confidence:** 3

**Review:**

The paper considers the Fourier spectrum of functions represented by Deep ReLU networks, as well as the relationship to the training procedure by which the network weights can be learned.

It is well-known (and somewhat obvious) that deep neural networks with rectifier activations represent piecewise linear continuous function. Thus, the function can be written as a sum of the products of  indicators of various polytopes (which define the partition of R^d) and the linear function on that polytope. This allows the authors to compute the Fourier transform (cf. Thm. 1) and the magnitude of f(k) decays as k^{-i} where the i can depend on the polytope in some intricate fashion. Despite the remarks at the end of Thm 1, I found the result hard to interpret and relate to the rest of the paper. The appearance of N_f in the numerator (which can be exponentially large in the depth) may well make these bounds meaningless for any networks that are relevant in practice.

The main paper only has experiments on some synthetic data.

Sec 3: Does the MSE actually go to 0 in these experiments? Or are you observing that GD fits lower frequencies, because it has a hard time fitting things that oscillate frequently?

Sec 4: I would have liked to see a clearer explanation for example of why increasing L is better for regression, but not for classification. As it stands I can't read much from these experiments.

Overall, I feel that there might be some interesting ideas in this paper, but the way it's currently written, I found it very hard to get a good "picture" of what the authors want to convey.

---

> ### Author Response · Authors · 2018-11-12
> **Response to Reviewer 2**
>
> Thank you for your constructive feedback!
>
> **On clarity of the message**
>
> Thanks for this valuable feedback. We understand your concerns about clarity. To that end, we have reworked parts of our exposition,  including the abstract and introduction, in the new revision, to make the message more clear.
>
> We believe our work follows lines of research on  the learning biases of neural networks, motivated by a lack of theoretical  understanding of the good generalization performance of these models despite their large capacity [Zhang et al. 2017]. We approach this through the lens of Fourier analysis, which we think is an original and interesting angle. Our main contribution is to show a learning bias of neural networks towards low frequency functions.  We believe this might be an important insight towards explaining why neural networks tend to prioritize learning simple patterns that generalize across data samples [Arpit et al. 2017].  We  also investigate the subtle interplay between the learnability of large frequencies and the geometry of the data manifold, pointing that a low frequency function in input space can fit large frequencies on highly curved manifolds.
>
> We hope this clarifies the picture we want to convey.
>
> **On the significance of Theorem 1:**
>
> Despite the analysis of Section 2.2 being of interest in its own right as a result on the spectral properties of ReLU networks,  Theorem 1 plays a central role in developing a formal understanding of the results in Sections 3 and 5. We have revised Section 3 in the manuscript to make the link to Theorem 1 more explicit.
>
> Although we choose to present the main result of Section 2.2 in the form of an asymptotic bound in Theorem 1, note that our analysis (Lemma 1 together with the procedure described in Diaz et al. 2016 and explained in detail Appendix D.2) actually allows us to get the Fourier components in closed form, for a given set of weight matrices. These components typically decay as fast as $k^{-d-1}$ where d is the input data dimension (around 1000 (!) for small-scale real-world problems (e.g. 784 for MNIST)) leading to larger contribution from lower frequencies relative to the higher ones.
>
> While the number $N_f$ of linear regions can be indeed be large, it affects all frequencies uniformly, i.e. leaves the spectral decay rate intact. Moreover, in most practical settings one usually constrains the Lipschitz constant $L_f$ (which appears together with $N_f$ in the numerator of the bound), e.g. with weight decay, batch norm [cf. Santurkar et al. 2018], gradient penalty, spectral normalization [Miyato et al. 2018], etc.
>
> **On experimenting with synthetic data**:
>
> While we understand and appreciate the need for showing consequences of our analysis on real data, our rationale for exclusively using synthetic data in the main text is that it affords us rich control over experimental parameters (e.g. shape of the manifold, frequency of functions defined on manifold). In a sense, it allows us to study the "raw" behaviour of the network, unconfounded by unknown external factors that might depend on the data in uncontrollable ways.
>
> **On the MSE Going to Zero**
>
> In Experiment 1 (Section 3), the mean squared error loss drops reasonably close to zero (typically around 0.05; the revision includes loss curves in the appendix).
>
> > GD fits lower frequencies, because it has a hard time fitting things that oscillate frequently?
>
> Your intuition is correct: it is indeed true and one of the central themes of the paper that high-frequency components of the target function (i.e. parts of the function that oscillate frequenctly) are harder to fit. If the target function contains extremely large frequencies, the convergence can be extremely slow.
>
> **On regression v.s classification results:**
>
> Our report of the results might not have been clear enough: increasing L is better for *both* regression and classification. You can observe in Figure 4 that increasing L (going up a column) yields better classification accuracies.
>
> **In closing:**
>
> We hope that our answer and revision make the main message of the paper more apparent.
>
> [Santurkar et al. 2018] https://arxiv.org/abs/1805.11604
> [Miyato et al. 2018] https://arxiv.org/abs/1802.05957
> [Zhang et al 2017] https://arxiv.org/abs/1611.03530
> [Arpit at al 2017] https://arxiv.org/abs/1706.05394

---

### Official Review · AnonReviewer3 · 2018-11-05
**Intriguing topic and analysis, but its impact on understanding of neural nets seems limited**

**Rating:** 6
**Confidence:** 3

**Review:**

Synopsis:
This paper analyzes deep Relu neural networks based on the Fourier decomposition of their input-output map. They show theoretically that the decomposition is biased towards low frequencies and give some support that low frequency components of a function are learned earlier under gradient descent training.

Pros:
--Fourier decomposition is an important and (to the best of my knowledge) mostly original angle from which the authors analyze the input-output map governing neural networks. There is some neat mathematical analysis contained here based off of the piecewise-linearity of deep Relu nets and Fourier decomposition of polytopes in input space.

--The setup in the toy experiments of Sec. 4 seems novel & thoughtful; the authors consider a lower-dimensional manifold embedded in a higher dimensional input space, and the Fourier decomposition of the composition of two functions is related to the decomposition of constituents.

Cons:
--While this paper does a fairly good job establishing that NNs are spectrally biased towards low frequencies, I’m skeptical of its impact on our understanding of deep neural nets. Specifically, at a qualitative level it doesn’t seem very surprising: intuitively (as the authors write in Sec. 5), capturing higher frequencies in a function requires more fine tuning of the parameters.  At initialization, we don’t have such fine tuning (e.g. weights/biases drawn i.i.d Normal), and upon training it takes a certain amount of optimization time before we obtain greater “fine tuning.” At a quantitative level, these results would be more useful if (i) some insight could be gleaned from their dependence on the architectural choices of the network (in particular, depth) or (ii) some insight could be gained from how the spectral bias compares between deep NNs and other models (as is discussed briefly in the appendix -- for instance, kernel machines and K-NN classifiers). The primary dependence in the spectral decay (Theorem 1) seems to be that it (i) decays in a way which depends on the input dimensionality in most directions and (ii) it is highly anisotropic and decays more slowly in specific directions. The depth dependence seems to arise from the constants in the bound in Theorem 1 (see my comment below on the bound).

--Relying on the growth of the weight norm to justify the network's bias towards learning lower frequencies earlier in training seems a bit tenuous to me. (I think the stronger evidence for learning lower frequencies comes from the experiments.) In particular, I'm not sure I would use the bound in Theorem 1 to conclude what would happen to actual Fourier components during training, since the bound may be far from being met. For instance, (1) the number of linear regions N_f changes during training -- what effect would this have? Also, (2) what if one were to use orthogonal weight matrices for training? Presumably the network would still train and generalize but the conclusions might be different (e.g. the idea that growth of weight norms is the cause of learning low frequency components earlier).

Miscellaneous:
--Would appreciate a greater discussion on the role of the cost function (MSE vs cross-entropy) in the analysis or experiments. Are the empirical conclusions mostly identical?

---

> ### Author Response · Authors · 2018-11-12
> **Response to Reviewer 3**
>
> Thank you for your thoughtful comments!
>
> **On the dependence on architectural choice**
>
> Following your suggestion, we have included a suite of qualitative ablation experiments demonstrating the effect of width, depth and max-norm in Appendix A.3 of the new revision. As anticipated from Theorem 1, we find that increasing depth indeed helps towards fitting high frequencies, more so than increasing width. Further, increasing the weight clip also has the same effect.
>
> **On the theoretical evidence of the learning bias**
>
> Thank you for this feedback. We completely agree that relying on the increasing spectral norm to explain the spectral bias (i.e. low frequencies learned first) can be restrictive. In the latest revision, we reinforce the theoretical argument by showing that the gradient of the MSE loss (w.r.t network parameters) inherits the spectral decay rate of the network function itself (Eq 11). Consequently, the residual (difference between function and target) at lower frequencies is weighted stronger than at higher frequencies.
>
> Regarding tightness of the bound:  note that although we choose to present the main result of Section 2.2 in the form of an asymptotic bound in Theorem 1, our analysis (Lemma 1 together with the procedure described in Diaz et al. 2016 and explained in detail Appendix D.2) actually allows us to get the Fourier components in closed form, for a given set of weight matrices. These components typically decay as fast as $k^{-d-1}$ where d is the input data dimension (around 1000 (!) for small-scale real-world problems (e.g. 784 for MNIST)) leading to larger contribution from lower frequencies relative to the higher ones.
>
> On the number $N_f$ of linear regions: During training, i.e. for a given architecture, we note that Raghu et al. 2016 provide tight upper-bounds for $N_f$ which depend on the width and depth of the network, along with the input dimensionality. The effect of $N_f$ is therefore somewhat limited (in contrast with $L_f$, which can become arbitrarily large as training progresses).
>
> **On the role of the cost function**
>
> Thank you for bringing up this interesting point! Note the brief discussion of the role of the MSE loss in Section 3 (Eq 10), showing that it induces no structural bias towards any particular frequency component (there is no weight coefficient $w(k)$ before $|f(k) - \lambda(k)|^2$ in  Eq. 10). This allows us to eliminate the loss function as a potential confounding factor when empirically demonstrating the spectral bias. The same cannot be said of the cross-entropy loss, which could potentially introduce additional biases and thereby make it difficult to isolate the bias due to the network parameterization from that due to the loss function itself. This is precisely why we used MSE in most of our experiments. We make this clearer in the latest revision.
>
> Note however that we did use cross entropy in Experiment 3, which reproduces Experiment 2 in the context of classification. We obtained similar results.
>
> We hope you find that our revision and clarifications address your concerns.

---

> > ### Comment · AnonReviewer3 · 2018-11-27
> > **Reviewer Reply**
> >
> > Thank you for the revisions and clarifications!
> >
> > Regarding architectural choices:
> > I appreciate the authors' inclusion of the ablation experiments! I do think it would have been nice to have more extensive experiments of this kind as a prominent focus of the paper.
> >
> > Regarding the effect of learning:
> > Thanks, I think analyzing the bias of the MSE gradient is a nice argument.
> >
> > Regarding the choice of cost function:
> > While it would have been interesting to have included analysis of cross-entropy in this paper, I think restriction to MSE is still suitable for scope.

---

### Official Review · AnonReviewer4 · 2018-11-12
**theoretical and empirical analysis of implicit bias in neural networks via Fourier coefficients.**

**Rating:** 4
**Confidence:** 4

**Review:**

Summary.

This paper has theoretical and empirical contributions on topic of Fourier coefficients of neural networks.  First is upper bound on Fourier coefficients in terms of number of affine pieces and Lipschitz constant of network.  Second is collection of synthetic data and trained networks whereupon is argued that neural networks focus early effort upon low Fourier coefficients.


Brief evaluation.

Pros:

+ This paper attacks important and timely topic: identifying and analyzing implicit bias of neural networks paired with standard training methods.

Cons:

- "Implicit bias" hypothesis has been put forth by many authors for many years, and this paper does not provide compelling argument that Fourier coefficients provide good characterization of this bias.

- Regarding "many authors for many years", this paper fails to cite and utilize vast body of prior work, as detailed below.

- Main theorem here is loose upper bound primarily derived from prior work, and no lower bounds are given.  Prior work does assess lower bounds.

- Experiments are on synthetic data; prior work on implicit regularization does check real data.


Detailed evaluation.

* "Implicit bias" hypothesis appears in many places, for instance in work of Nati Srebro and colleagues ("The Implicit Bias of Gradient Descent on Separable Data" (and follow-ups), "Exploring generalization in deep learning" (and follow-ups), and others); it can also be found in variety of recent generalization papers, for instance again the work of Srebro et al, but also Bartlett et al, Arora et al.  E.g., Arora et al do detailed analysis of favorable biases in order to obtain refined generalization bound.  Consequently I expect this paper to argue to me, with strong theorems and experiments, that Fourier coefficients are a good way to assess implicit bias.

* Theorem 1 is proved via bounds and tools on the Fourier spectra of indicators of polytopes due to Diaz et al, and linearity of the Fourier transform.  It is only upper bound (indeed one that makes no effort to deal with cancellations and thus become tight).  By contrast, the original proofs of depth separation for neural networks (e.g., Eldan and Shamir, or Telgarsky, both 2015), provide lower bounds and metric space separation.  Indeed, the work of Eldan&Shamir extensively uses Fourier analysis, and the proof develops a refined understanding of why it is hard for a ReLU network to approximate a Fourier transform of even simple functions: it has to approximate exponentially many tubes in Fourier space, which it can only do with exponentially many pieces.  While the present paper aims to cover some material not in Eldan&Shamir --- e.g., the bias with training --- this latter contribution is argued via synthetic data, and overall I feel the present work does not meet the (high) bar set by Eldan&Shamir.

*  I will also point out that prior work of Barron, his "superposition" paper from 1993, is not cited. That paper presents upper bounds on approximation with neural networks which depends on the Fourier transform.  There is also follow-up by Arora et al with "Barron functions".

* For experiments, I would really like to see experiment showing Fourier coefficients at various stages of training of standard network on standard data and standard data but with randomized labels (or different complexity in some other way).  These Fourier coefficients could also be compared to other "implicit bias" quantities; e.g., various norms and complexity measures.  In this way, it would be demonstrated that (a) spectral bias happens in practice, (b) spectral bias is a good way of measuring implicit bias.  Admittedly, this is computationally expensive experiment.

* Regarding my claim that Theorem 1 is "loose upper bound": the slope of each piece is being upper bounded by Lipschitz constant, which will be far off in most regions.  Meanwhile, Lemma 1, "exact characterization", does not give any sense of how the slopes relate to weights of network.  Improving either issue would need to deal with "cancellations" I mention, and this is where it is hard to get upper and lower bounds to match.

I feel this paper could be made much stronger by carefully using the results of all this prior work; these are not merely citation omissions, but indeed there is good understanding and progress in these papers.

---

> ### Author Response · Authors · 2018-11-15
> **Response to Reviewer 4 [2/2]**
>
> Part [2/2]
>
> **Regarding Tightness of Bound:**
>
> For some context: the primary motivation behind Section 2 is to develop a formal framework for understanding of the results in Sections 3 and 5 - not to derive approximation error bounds. In a recent revision, we have updated Section 3 in the manuscript to make the link to Theorem 1 more explicit. For example, we now show that the gradient of the MSE loss (w.r.t network parameters) inherits the spectral decay rate of the network function (Eq 11). Consequently, the residual (difference between function and target) at lower frequencies is weighted stronger than at higher frequencies. Folllowing a suggestion of AnonReviewer3, we also included a suite of qualitative ablation experiments demonstrating the effect of width, depth and max-norm in Appendix A.3 of the new revision. As anticipated from Theorem 1, we find that increasing depth indeed helps towards fitting high frequencies, more so than increasing width. Further, increasing the weight clip also has the same effect.
>
> On tightness: The Fourier coefficients take the general form of a rational, typically homogeneous  function $C(W_\epsilon, \hat k) / k^{-d-1}$.
>
> (1) The actual inequality originates from the terms C  depending on linear mappings operating on weights $W_{\epsilon}$ and general unit vectors $\hat{k}$, as can be seen by recursively expanding the FT of the polytope as in Diaz et al. 2016. We think that the requirement of generality leaves little scope for further cancellations and tighter bounds.
>
> In other words, for a general $k$, the tightness of the bound depends on the weight matrices. Indeed, we provide empirical evidence in Appendix A.3: increasing the weight clip (i.e. by relaxing the upper bound on the parameter max-norm, and by proxy on the Lipschitz constant) has a significant impact towards whether the network can match the target function in the Fourier domain. This implies that in this particular setting, the bound must be tight, given that it is preventing the network from learning higher frequencies.
>
> (2) Observe that in equation 11, the inequality can only affect all frequencies uniformly. In other words, the scaling behaviour of the fourier coefficients with increasing $k$ remains intact, irrespective of the numerator. Therefore, the down-scaling of the contribution towards the loss gradient of the residual at higher frequencies is not affected by the tightness of the bound.
>
> **Regarding Synthetic Data**
>
> While we understand and appreciate the need for showing consequences of our analysis on real data (in Appendix), our rationale for exclusively using synthetic data in the main text is that it affords us rich control over experimental parameters (e.g. shape of the manifold, frequency of functions defined on manifold). In a sense, it allows us to study the "raw" behaviour of the network, unconfounded by unknown external factors that might depend on the data in uncontrollable ways.
>
> **In closing**
>
> We hope our response and the updated revision address your concerns.

---

> ### Author Response · Authors · 2018-11-15
> **Response to Reviewer 4 [1/2]**
>
> Thank you for your thorough comments and feedback.
>
> Our response is split in two parts.
>
> Part [1/2]
>
> **Context of the paper and related work**
>
> Thank you for pointing out the importance of the topic! It is also our view that our work should be understood in the context of the very active lines of research on expressivity and implicit bias in neural networks.
>
> To allow for a fair comparison of our work with the existing literature, we feel it is worth clarifying what is specifically  addressed in our paper - and what is not:
>
> 1. What we do (our novelty):
> (i) We describe a computation procedure for the Fourier transform and spectrum of deep ReLU networks. To our knowledge, although the analysis is largely inspired by techniques developed by [Diaz et al. 2016] to evaluate the Fourier shape transform of polytopes, this is a new result. (ii) We build upon this result to show a learning bias of neural network towards low frequency functions. This is motivated by the recent observation made in [Arpit et al. 2017] that neural networks tend to prioritize learning simple patterns that generalize across data samples. (iii) We investigate the subtle interplay between the learnability of large frequencies and the geometry of the data manifold. We believe this a novel and original insight.
>
> 2. What we don't do:
> (i) Our work departs from analysis of approximation bounds. This is not our goal.
> (ii) Although we believe this is an important and challenging problem, we do not aim at a full characterization of the implicit bias of gradient descent. This requires to tackle the learning dynamics of non-linear neural networks, which is, to the best of our knowledge, a largely open research topic  -- not (directly) addressed in our paper.
> (iii) Our goal is not to derive generalization bounds.
>
> **On the related work on Implicit Bias:**
>
> Thank you for the nice references! We have included some of them in our latest revision: both in the introduction,  to make the context of our work more explicit; and in the Related Work section that has been expanded accordingly. We note however that having strong theorems in the context of largely intractable systems such as neural networks often requires making strong assumptions. For instance, the main theorem of the reference Soudry et al,  "The Implicit Bias of Gradient Descent on Separable Data", concerns logistic regression on linearly separable data!
>
> We would also like to point out that the term "implicit bias" is quite generic and can have many interpretations. In our work, we consider learning bias in the ubiquitous class of deep ReLU networks. We believe this is an important first step for future work to build on, given how we expose in a principled manner that a learning  bias in the Fourier domain indeed exists for deep ReLU networks.
>
> **On the related work on expressivity:**
>
> Thank you for pointing out the missing references on architecture-dependent approximation bounds;  we have included them in our new revision. We feel there may have been a confusion about the purpose of Theorem 1: it should not be understood as an approximation bound. The goal in Section 2 is  to compute the Fourier transform; we choose to present the main result of Section 2.2 in the form of an asymptotic bound, which puts the emphasis on the spectral decay. We discuss more the significance of Theorem 1 below.
>
> On [Eldan & Shamir 2015]:
>
> Thank you for this nice reference! This paper makes elegant use of specific properties of the Fourier transform of 2-layer networks to show a specific depth-separation result.  Although the motivation for our paper differs from theirs, their proof indeed gives a nice insight on why 2-layer networks do not approximate high frequency functions well. We have included this in the Related Work section. Note however that:
> (i) The two papers address entirely different questions: our primary goal is to expose the spectral bias of deep ReLU networks during learning, while their goal is to illustrate the role of depth in expressivity, through a worse-case separation analysis between 2 and 3-layer networks.
> (ii) Furthermore, the techniques developed in their paper is of little help for our goal: (a) they do not actually compute the Fourier coefficients and (b) their argument is tailored towards 2-layer networks only (i.e. those having the form $\sum_i f_i(<v_i,x>)$ for some activation function $f$).
>
> [Continued in Part 2]

---

> > ### Comment · AnonReviewer4 · 2018-11-20
> > **thanks**
> >
> > Thank you for your comments.
> >
> > Thank you for looking over prior work I mentioned.  You may crawl google scholar from these to find more.
> >
> > Unfortunately, you have not moved me away from my core assessment.  Succinctly:
> >
> > 1. Paper does not provide me with compelling argument that this spectral bias exists; as I said, Theorem 1 is loose, and experiments are on synthetic data.
> >
> > Explaining further:
> >
> > 1. Based on your paragraph starting with "1. What we do (our novelty)", I feel I have not misrepresented your contribution; namely, (i) is Theorem 1, and (ii) and (iii) are based on synthetic experiments.  The theorem is only an upper bound so it can easily happen that Fourier spectrum is much different.  Similarly, experiments are stylized synthetic data.  Consequently, on real data, trend may be completely different, even reversed.  Indeed, Fourier transforms work best on smooth functions.  For deep networks, which are not only piecewise affine but (e.g., due to existence of adversarial examples) have highly nonsmooth regions, Fourier transform can be expected to be messy.  There could easily be many learning problems that seem well-behaved in every possible way, yet Fourier approach advocated here suggests otherwise.  From what is provided in this paper, I simply do not know, and am not compelled to use Fourier coefficients to study implicit bias of deep networks.  I feel prior work I mentioned sets much higher bar in terms of what is possible and needed to make compelling argument (due to this as well, i am not compelled to increase score based purely on technical contribution of Theorem 1).

---

> > > ### Author Response · Authors · 2018-11-21
> > > **Author Response**
> > >
> > > Thank you for this exchange.
> > >
> > >  > The theorem is only an upper bound so it can easily happen that Fourier spectrum is much different.
> > >
> > > We reiterate that the analysis of Section 2 does not merely give a bound, but the closed-form expression for the Fourier transform (Lemma 1 and Eq 27) of deep ReLU networks. We have added Diaz’ formula for the Fourier transform of polytopes in Eq 27 of Appendix D.2.2 to make this point even more explicit. We obtain the Fourier coefficients as  rational functions of k,  giving the anisotropic spectral decay spelled out in Theorem 1.
> > >
> > > > Paper does not provide me with compelling argument that this spectral bias exists; as I said, Theorem 1 is loose
> > >
> > > The analysis of Section 2 is agnostic of the learning method; so obviously, a compelling argument that a learning bias exists needs additional justification. However, as already mentioned in our previous reply, the learning bias does not emerge from Theorem 1 independently, and is certainly not contingent on the (tightness of the) bound. To reinforce the theoretical argument about the learning bias, we show in Section 3 (Eq 11) that the gradient of the MSE loss (w.r.t network parameters) inherits the spectral decay rate of the network and hence gradient descent based methods will admit such bias. As for lemma 1 and Theorem 1, they reveal the structure of the spectrum of ReLU networks which in itself is insightful, including the dependence of the bound on the architecture whose tightness we probe experimentally in Appendix A.3.
> > >
> > > > experiments are on synthetic data.
> > >
> > > Note that we are not the first to identify that there is a learning bias in deep ReLU networks. Extensive experiments on real data that support the learning bias claim have already been done in Arpit et al (2017). Our contribution is to formalize this observation using the framework of Fourier transform. We would also like to clarify the scope of our experiments: they are an integral part of the analysis. Their purpose is not merely to validate theoretical results, but to guide the pursuit of deeper theoretical insights (see discussions after experiments 1, 3 and 4).
> > >
> > > We carry out these experiments in a controlled setting to minimize the effect of unknown and potentially misleading confounding factors. Like we emphasized in Part 2 of our previous answer: unlike real data, synthetic data affords us such control (e.g. over the shape of the data-manifold, the target function, the sampling distribution, etc). While we do indeed show consequences of our analysis on real data (Cifar10 and MNIST) in Appendix A4 and A5, these experiments are meant to supplement the empirical analysis of the main paper – not to replace it.
> > >
> > > Finally, we note that many seminal papers make extensive use of synthetic experiments (e.g. Poole et al. 2016); in fact, many significant theoretical contributions make strong simplifying assumptions about the model (e.g Soudry et al. 2016) and are not constructed to model realistic datasets (e.g Eldan and Shamir 2015 – the worse-case data distribution in their proof corresponds to the indicator of a L2 ball in Fourier space). We believe our work provides new insights - both theoretical and empirical - without making any strong assumptions, and in a context (the learning dynamics of deep ReLU networks) where very little is known.
> > >
> > > > Consequently, on real data, trend may be completely different, even reversed.
> > >
> > > Besides the fact that we do show experiments on real data in the appendix, we feel there is no reason to expect such an anomalous behaviour on real data given that we do not make any assumption on the target function and data (except that it is on a bounded domain, which is true for images). In fact, in Arpit et al (2017), who conduct all their experiments on real data, it was shown that deep ReLU networks always prioritized learning low complexity functions.
> > >
> > > > For deep networks, which are not only piecewise affine but (e.g., due to existence of adversarial examples) have highly nonsmooth regions
> > >
> > > Please note that we are not claiming that deep networks are all smooth with only low frequency modes. What our analysis shows is that the learning is biased towards smooth functions, in the sense that it prioritizes low frequencies, which are learned first/faster.
> > >
> > > The existence of adversarial examples is not incompatible with our claims. In fact, we hope our work in Section 4 inspires future work towards understanding adversarial examples from the perspective of sensitivity analysis under the manifold hypothesis.
> > >
> > > > I feel prior work I mentioned sets much higher bar in terms of what is possible
> > >
> > > Like we pointed out in our previous reply,  the prior work you mentioned address specific problems that are different than ours, and in a different (and often simplified) context. Hence, we do not feel the comparison with our work is fair and relevant to assess our results.

---

> > > > ### Comment · AnonReviewer4 · 2018-12-02
> > > > **thank you**
> > > >
> > > > Thank you for your time and comments.
> > > >
> > > > I have updated my review to detail two issues:
> > > >
> > > > - Compelling experiments (estimating Fourier coefficients),
> > > >
> > > > - Looseness of bounds (dealing with cancellations).

---

> > > > > ### Author Response · Authors · 2018-12-02
> > > > > **Thank you for your additions.**
> > > > >
> > > > > Thank you for your additions. We feel they re-iterate concerns you had already made clear in the first version of your review, so it could be that we have been talking at cross-purposes.
> > > > >
> > > > > > Meanwhile, Lemma 1, "exact characterization", does not give any sense of how the slopes relate to weights of network.
> > > > >
> > > > > The relation between slope and weights is given by Eq 3. Details are given in Appendix C.
> > > > >
> > > > > > the slope of each piece is being upper bounded by Lipschitz constant, which will be far off in most regions [...] Improving either issue would need to deal with "cancellations" I mention
> > > > >
> > > > > To our current understanding, we have no reason to expect any 'cancellation' phenonenon to occur for general architectures.
> > > > >
> > > > > Please also consider our addition of ablation experiments in Appendix A.3. As we mentioned in a previous comment (https://openreview.net/forum?id=r1gR2sC9FX&noteId=SJlUNA4i6m ), increasing the Lipschitz constant has a significant impact towards whether the network can match a target function in the Fourier domain, implying that e.g. when regressing high frequency functions, the bound can indeed be tight.
> > > > >
> > > > > > I would really like to see experiment showing Fourier coefficients at various stages of training of standard network on standard data [...] Admittedly, this is computationally expensive experiment.
> > > > >
> > > > > Indeed. For example, with (say) 784 dimensional inputs (on MNIST), it requires evaluating the network on a dense 784 dimensional grid. Even if the grid has 100 points per dimension, that amounts to 100^784 forward passes through the network.

---

### Author Response · Authors · 2018-12-03
**Comment to All Reviewers**

We thank all reviewers for their feedback. Most importantly, in the discussions and revision,  we tried to clarify the contribution of Section 2 and its role in the paper.  We have revised  Appendix D.2.2 to make it more clear that we obtain a closed form expression for the Fourier transform.  We reinforced the theoretical argument for the spectral bias by analyzing the bias of the MSE gradient (Eq 11).  Following AnonReviewer3's suggestion, we have included a suite of ablation experiments (Appendix D.3), showing the effect of architecture and the Lipschitz constant on the spectral bias. We hope this improved clarity makes the significance of our contributions more apparent.

Main Contributions:

1. We describe a computation procedure for the Fourier transform and spectrum of deep ReLU networks.

2. We show a learning bias of neural networks towards low frequency functions.  We believe this might be an important insight towards explaining why neural networks tend to prioritize learning simple patterns that generalize across data samples [Arpit et al. 2017].

3. We investigate the subtle interplay between the learnability of large frequencies and the geometry of the data manifold, pointing that a low frequency function in input space can fit large frequencies on highly curved manifolds.

We believe our work provides new insights - both theoretical and empirical - without making any strong assumptions, and in a context (the learning dynamics of deep ReLU networks) where very little is known.

---

### Author Response · Authors · 2018-12-16
**Real Data Consequence of the Spectral Bias**

Although we can’t update our submission anymore: following the reviewers' suggestion, we have ran a few experiments on MNIST to demonstrate the effect of spectral bias on real data. It involves evaluating the robustness of neural network training dynamics to noise of various frequencies.

We train the same 6-layer deep 256-unit wide network on MNIST images of classes “1” and “7”. The ground-truth label is +/- 1 for the respective classes, and the network is trained with full-batch gradient descent with MSE loss and Adam. From the plots found here [1], we make the following observations.

(a) Adding a low frequency noise signal to labels degrades the generalization peformance (the difference between training and validation losses) to a much larger extent than high-frequency noise of the same amplitude.

(b) The network is instantly able to fit the low frequency noise signal, causing the validation performance (w.r.t. clean validation labels without noise) to suffer. The training loss is small, as expected. In particular, observe that the validation performance is quite sensitive to the amplitude of the noise – the only way to improve performance is by decreasing the noise amplitude.

(c) High-frequency noise is only fit later in the training. This results in a dip in the validation score: this is around when the true (low-frequency) labels are learned. As the training progresses, higher frequencies are fit, which results in increasing validation loss but decreasing training loss. Further, observe that the validation performance around the dip is fairly robust to change in noise amplitude – this is expected, since the amplitude of the high frequency noise shouldn’t affect the learning of the true target this early in the training.

[1] https://imgur.com/a/pyMfCiL

---

### Meta-Review · Area_Chair1 · 2018-12-11
**ICLR 2019 decision**

**Confidence:** 4
**Recommendation:** Reject

**Metareview:**

This paper considers an interesting hypothesis that ReLU networks are biased towards learning learn low frequency Fourier components, showing a spectral bias towards low frequency functions. The paper backs the hypothesis with theoretical results computing and bounding the Fourier coefficients of ReLU networks and experiments on synthetic datasets.

All reviewers find the topic to be interesting and important. However they find the results in the paper to be preliminary and not yet ready for publication.

On theoretical front, the paper characterizes the Fourier coefficients for a given piecewise linear region of a ReLU network. However the bounds on Fourier coefficients of the entire network in Theorem 1 seem weak as they depend on number of pieces (N_f) and max Lipschitz constant over all pieces (L_f), quantities that can easily be exponentially big.  Authors in their response have said that their bound on Fourier coefficients is tight. If so then the paper needs to discuss/prove why quantities N_f and L_f are expected to be small. Such a discussion will help reviewers in appreciating the theoretical contributions more.

On experimental front, the paper does not show spectral bias of networks trained over any real datasets. Reviewers are sympathetic to the challenge of evaluating Fourier coefficients of the network trained on real data sets, but the paper does not outline any potential approach to attack this problem.

I strongly suggest authors to address these reviewer concerns before next submission.